# RETHINKING BEHAVIOR REGULARIZATION IN OFFLINE SAFE RL: A REGION-BASED APPROACH

## ABSTRACT

Behavior regularization is a widely adopted technique in offline reinforcement learning (RL) to control distributional shift and mitigate extrapolation errors from out-of-distribution (OOD) actions by keeping the learned policy close to the behavior policy used to collect the dataset. However, directly applying behavior regularization to offline safe RL presents several issues. The optimal policy in safe RL should not only favor actions that prevent the agent from entering unsafe regions but also identify the shortest escape path when the agent finds itself in unsafe states. Enforcing safety and behavior regularization constraints simultaneously is inherently difficult and can often lead to infeasible solutions, especially when multiple constraints are involved. Furthermore, adding behavior regularization may cause the learned policy to imitate the behavior policy, even in states where the behavior policy performs poorly (not safe). This issue becomes particularly severe in offline safe RL, where the quality of the dataset collected by the behavior policy heavily impacts the learned policy's effectiveness. To address these challenges, we propose *BARS* (Behavior-Aware Region-Based Safe offline RL), a novel algorithm that distinguishes between safe and unsafe states and applies region-specific, selective behavior regularization to optimize the policy in safety-critical applications. Extensive experiments show that BARS significantly outperforms several state-of-the-art baselines in terms of both rewards and safety, particularly in scenarios where the behavior policy is far from optimal. Notably, when dataset quality is low, BARS continues to perform well and ensure safety, while all other baselines fail to guarantee a safe policy in most of the environments. Our work has great potential to address a previously overlooked issue in offline safe RL.

## 1 INTRODUCTION

Safe reinforcement learning (RL) has been successfully applied in areas such as autonomous driving (Isele et al., 2018), recommend systems (Chow et al., 2017), and robotics (Achiam et al., 2017; Dawson et al., 2023), helping to develop policies that adhere to critical safety constraints, including collision avoidance, budget management, and system reliability. However, a significant challenge lies in gathering the data needed to train these policies in an online setting. In many real-world applications, interacting with the environment is not only costly and time-consuming but also inherently risky. Training an RL agent online requires learning through trial and error, which can lead to unsafe behavior during the learning process—an unacceptable risk in safety-critical tasks.

To overcome the limitations of data collection, offline RL offers a safer alternative by enabling policies to be learned from pre-collected data, without the need for direct interaction with the environment during training. Instead of relying on real-time exploration, offline safe RL leverages datasets gathered from prior experiences, eliminating the risk of unsafe actions during the learning process. However, in offline RL, a major problem arises when the learned policy encounters out-of-distribution (OOD) actions (Fujimoto et al., 2019) since the policy is trained on a static, pre-collected dataset. Thus, the policy may attempt actions that were not well-presented during training, leading to extrapolation errors in the value function, which can not be eliminated without requiring coverage of the offline dataset. Existing works tackle this problem by restricting the learned policy to only select actions that have sufficient support under the behavior policy by adding a behavior regularization term in both offline RL (Peng et al., 2019; Siegel et al., 2020; Wang et al., 2020; Jaques et al., 2019; Wu et al., 2019; Dadashi et al., 2021) and offline safe RL (Lee et al., 2022; Xu et al., 2022; Lin et al.,

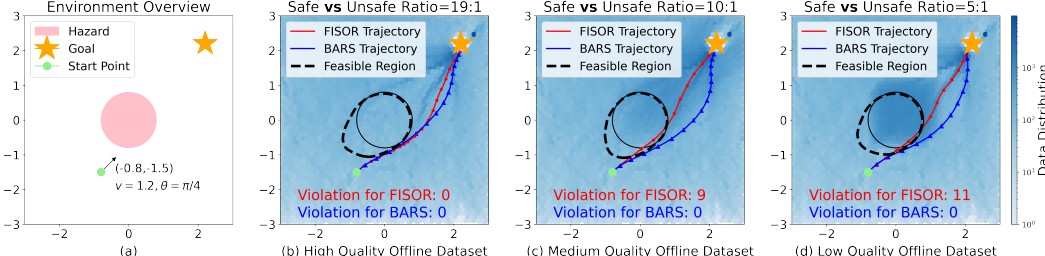

Figure 1: **(a)**: Reach-avoid task: The agent starts from a fixed point (green) and aims to reach the goal (orange) while avoiding hazard zones (pink). The initial state is set at $(-0.8, -1.5)$ with a velocity of $1.2$ and a movement direction of $\pi/4$ radians from the horizontal. Any state in the hazard region incurs a cost of $1$, while all other states have a cost of $0$. The total constraint violation is the number of steps spent in the hazard region. The goal is to learn an optimal policy that reaches the goal without violating constraints. **(b)**: High-quality offline data is generated using an expert policy trained by TD3-Lag modified on TD3 (Fujimoto et al., 2018). **(c)**: Medium-quality data is collected from a mixture of the expert policy in (b) and a random policy, resulting in a safe-to-unsafe data ratio of $10:1$. **(d)**: To increase unsafe data, more random policy data is gathered in the hazard region, leading to a safe-to-unsafe ratio of $5:1$. The "Feasible Region" represents the largest safe region. See Section 4 for more details. In all scenarios, BARS can successfully learn a **safe** and optimal policy across various data distributions, while FISOR fails to maintain safety as the data quality degrades.

2023; Guan et al., 2024; Zheng et al., 2024). While behavior regularization has proven effective for managing distributional shift in standard offline reinforcement learning, its direct application to offline safe RL introduces significant challenges. Safe RL is typically modeled as a Constrained Markov Decision Process (CMDP), where the goal is to maximize rewards while satisfying safety constraints. In some CMDPs, certain states are so high-risk that no truly safe actions or trajectories are available in those states.. In such cases, the optimal solution is to select the action with the minimum cost. For example, imagine an autonomous vehicle experiencing a brake malfunction. Every available action poses a risk—stopping abruptly could cause an accident, turning might result in a skid, and continuing forward could lead to a more dangerous situation. Here, the vehicle must choose the least harmful action, illustrating the need to navigate risk effectively when no safe options are available. Therefore, the optimal policy in safe RL must not only avoid unsafe regions but also find the quickest escape route when forced to enter an unsafe state. Enforcing safety constraints alongside behavior regularization often leads to conflicting objectives, making it difficult to find feasible solutions.

Another key drawback of applying behavior regularization directly in offline safe RL is that it may cause the learned policy to overly imitate the behavior policy, even in states where the behavior policy performs poorly or unsafely. Since offline RL relies on pre-collected datasets, the quality of the behavior policy directly impacts the learned policy's effectiveness. If the dataset contains sub-optimal or unsafe behavior, behavior regularization may reinforce these undesirable actions, resulting in poor performance. This issue is particularly severe in safety-critical environments.

To address these challenges, we propose *BARS* (Behavior-Aware Region-Based Safe offline RL), a novel algorithm that distinguishes between safe and unsafe states while applying region-specific, selective behavior regularization to optimize the policy. As visualized in Figure 1, we consider a simple toy example, where the agent aims to find the shortest path from a fixed starting point to a destination while avoiding hazardous zones. The performance of the behavior policy is regulated by controlling the ratio between an expert and a random policy used to collect the offline dataset. The figure demonstrates that BARS consistently performs well in different scenarios (high/medium/low-quality offline dataset), while the state-of-the-art algorithm FISOR (Zheng et al., 2024) fails to satisfy the safety constraints as the dataset quality degrades. More discussions on the related work can be found in Section A in the Appendix. Our contributions are summarized as follows:

- BARS is the first to address the issue of behavior regularization constraints leading to infeasible solutions in safe RL using a region-based approach. Infeasible solutions are especially likely to occur when multiple constraints are involved. Unlike traditional methods, BARS adapts regularization based on state safety (Eq.(4)), maximizing rewards in safe regions and minimizing risks in unsafe regions by selecting the least harmful actions.

- We propose a "Region-Based Risk-Reward Optimization" (Eq.(7)) in our algorithm, which handles value function estimation errors and penalizes unsafe actions when optimizing in safe regions.

- Extensive experiments on the DSRL benchmark (Liu et al., 2023a) show that BARS significantly outperforms state-of-the-art baselines in reward maximization and constraint satisfaction. On Safety-Gymnasium tasks (Ray et al., 2019; Ji et al., 2023), BARS increases rewards by **5%** and reduces costs by **52%** on average compared to the best baseline (Table 1). Additionally, Table 2 shows BARS consistently maintains safety as the data ratio varies. Ablation studies evaluate its performance under different cost limits.

- To explore how to **safely** learn a safe policy, we propose a train-collect-and-train framework, which significantly enhances BARS's performance when new data collection is allowed. With just one round of new data, performance improved by **72%**, with the new dataset ensuring policy safety.

## 2 PRELIMINARY

We formulate safe-RL as a Constrained Markov Decision Process (CMDP), denoted by the tuple $\mathcal{M} = (\mathcal{S}, \mathcal{A}, \gamma, \mathcal{P}, r, c, \rho)$. Here, $\mathcal{S}$ and $\mathcal{A}$ represent the state and action spaces, $\rho$ is the initial state distribution, $\gamma \in [0,1)$ is the discount factor, $r : \mathcal{S} \times \mathcal{A} \to \mathbb{R}$ is the reward function, and $c : \mathcal{S} \times \mathcal{A} \to \mathbb{R}^+$ is the cost function. WLOG, we use $c = 0$ to indicate that the state constraint is satisfied (the state is safe). At each time step, an agent observes a state $s \in \mathcal{S}$, takes an action $a \in \mathcal{A}$ according to a policy $\pi$, receives a reward $r(s,a)$ and a cost $c(s,a)$, and then transitions to the next state $s'$ based on the transition kernel $\mathcal{P}(\cdot|s,a)$. Given a policy $\pi$, we use $V_r^\pi(s)$ and $V_c^\pi(s)$ to denote the expected cumulative discounted return and cost under policy $\pi$, starting from state $s$: $V_r^\pi(s) = \mathbb{E}\left[\sum_{t=0}^\infty \gamma^t r(s_t, a_t)|s_0 = s\right], V_c^\pi(s) = \mathbb{E}\left[\sum_{t=0}^\infty \gamma^t c(s_t, a_t)|s_0 = s\right]$. It is well-known that the optimal policy of a discounted MDP (or CMDP) (Sutton and Barto, 2018; Altman, 1999; Bertsekas, 2019) depends on the initial state. To better illustrate the idea and address existing issues in offline safe RL, we consider a fixed initial state $\rho = s$ in this paper. The objective for safe-RL is to find a policy $\pi : \mathcal{S} \to \Delta(\mathcal{A})$ ($\Delta(\cdot)$ is a probability simplex) to maximize the expected cumulative rewards while satisfying constraints:

$$\max_\pi V_r^\pi(s) \quad \text{s.t.} \quad V_c^\pi(s) \le l, \tag{1}$$

where $l$ represents the cost limit. Note that the cost limit specifies how many violations are allowed starting from a state $s$, as any positive value contributes to a violation based on the definition of the cost function. In other words, we do not permit cost compensation. In offline RL, we assume access to an available offline dataset $\mathcal{D} = \{(s_i, a_i, r_i, c_i, s_i')\}_{i=1}^N$, which consists of $N$ samples containing both safe and unsafe trajectories, collected using some underlying behavior policy $\pi_\beta$.

### 2.1 LIMITATIONS OF EXISTING SOLUTIONS

To handle safety constraints, existing approaches often consider solving the following optimization problem:

$$\max_\pi V_r^\pi(s) \quad \text{s.t.} \quad V_c^\pi(s) \le l, \ D(\pi||\pi_\beta) \le \epsilon, \tag{2}$$

where $\epsilon > 0$, $D$ is some divergence metrics (e.g., KL divergence) used for measuring the distance between the policy $\pi$ and behavior policy $\pi_\beta$ to prevent the distributional shift from $\pi_\beta$ in offline setting (Lee et al., 2022; Xu et al., 2022; Lin et al., 2023; Zheng et al., 2024). However, this formulation has several issues. 1). The policy to be optimized aims to maximize the accumulated reward while maintaining safety in an expected sense. In the unconstrained case, this approach is well justified, as all states are considered safe. However, in the safe-RL setting, particularly for safety-critical applications, ensuring safety on average across all states may overlook significant risks in a few critical states where $V_c^\pi(s) > l$. These harmful states, though rare, can still cause substantial damage. For example, in autonomous driving, the system may perform well in common scenarios such as highway driving or empty streets. However, in rare but critical situations like navigating a busy intersection or avoiding a pedestrian, the risks are much higher. Simply ensuring safety on average may overlook these high-risk states, where poor decisions can lead to serious accidents. 2), the two constraints $V_c^\pi(s) \le l$ and $D(\pi||\pi_\beta) \le \epsilon$ in the optimization problem (2) may not always be satisfied simultaneously, which means that there is no feasible solution. This issue becomes

more severe when multiple constraints involving different sets of cost functions are considered. 3) Enforcing the constraint to control distributional shift may result in a policy that is too close to the behavior policy for all the states, implying that the learned policy could perform as poorly as the behavior policy in states where the latter performs badly. This problem is more serious in the offline safe-RL setting, where an algorithm is trained on a dataset collected by the behavior policy, making the performance of the learned policy heavily dependent on the quality of the dataset (as shown in Figure.1).

Consider a simple MDP (Figure 2) with the following deterministic reward and cost functions defined for the states: $s_1, s_3, s_5$ have reward $r = 1$ and cost $c = 0$, while $s_2$ and $s_4$ have reward $r = 1$ and cost $c = 1$. The behavior policy $\pi_\beta$ is defined as $\pi_\beta(s_2) = a_5$, meaning action $a_5$ is chosen at $s_2$. Our goal is to learn an optimal policy $\pi^*$ that reaches $s_5$ with minimal cost violations. Using the total variation divergence as the divergence metric, we have $D(\pi||\pi_\beta) = 1 - p$, assuming $\pi(a_5|s_2) = p$. First, if we consider $s_2$ as the starting point, it is clear that the standard optimization problem (2) has no solution when $0 < \epsilon < 1$ and the cost limit $l = 1$, as it requires $p \geq 1 - \epsilon > 0$,

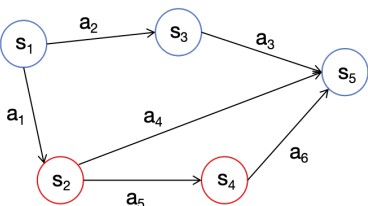

Figure 2: A MDP example

and $V_c^\pi(s_2) = 1 + p * V_c^\pi(s_4) > 1 + p$. Secondly, while the optimization problem (2) may have a solution for a larger cost limit $l$, setting $\epsilon$ small enough forces the policy $\pi$ to choose $a_5$ with a very high probability, $p \geq 1 - \epsilon$, which approaches one. As a result, the agent never learns the optimal action $a_4$. Thus, it is crucial to develop an approach that can distinguish between "good" states—where the policy satisfies the constraints—and "bad" states—where the policy fails to satisfy the constraints—and apply different training strategies to address the issues of infeasible solutions and poor data quality.

## 3 METHODS

To address these challenges, we first separate the entire state space into *safe region* ($\mathcal{S}_{\text{safe}}^\pi$) and *unsafe region* ($\mathcal{S}_{\text{unsafe}}^\pi$), which is defined below:

**Definition 3.1** *The safe region under a policy $\pi$ is defined as $\mathcal{S}_{safe}^\pi := \{s|V_c^\pi(s) \leq l\}$, and the unsafe region is $\mathcal{S}_{unsafe}^\pi := S \setminus \mathcal{S}_{safe}^\pi$.*

Then we aim to solve the following problem with separate states into consideration:

$$\textbf{If } s \in \mathcal{S}_{\text{safe}}^\pi : \max_\pi V_r^\pi(s) \qquad\qquad \textbf{If } s \in \mathcal{S}_{\text{unsafe}}^\pi : \max_\pi -V_c^\pi(s)$$
$$\text{s.t.} \quad V_c^\pi(s) \leq l, \qquad\qquad\qquad \text{s.t.} \quad D(\pi||\pi_\beta) \leq \epsilon, \qquad (3)$$
$$D(\pi||\pi_\beta) \leq \epsilon,$$

The optimization problem (3) distinguishes between states in safe regions and those in unsafe regions, solving them separately. For states within the safe region, the objective is to maximize rewards while adhering to the safety constraints. For states in the unsafe region, the focus shifts to minimizing future constraint violations as much as possible, as the rewards are meaningless when the agent is in the unsafe region. Although the optimal solution to the optimization problem (3) ensures a high reward while satisfying the constraints in the safe region and minimizes costs in the unsafe region (by escaping as quickly as possible), if a solution exists, it does not address issues (2) and (3) mentioned in Section 2.1. Additionally, learning the value function to determine the safe region for a policy $\pi$ requires Monte Carlo estimation through interactions with the environment, which is not feasible in an offline RL setting. To overcome these challenges, we propose the following optimization problem:

$$\textbf{If } s \in \mathcal{S}_{\text{safe}} : \max_\pi V_r^\pi(s) \qquad\qquad \textbf{If } s \in \mathcal{S}_{\text{unsafe}} : \max_\pi -V_c^\pi(s)$$
$$\text{s.t.} \quad V_c^\pi(s) \leq l, \bar{D}(\pi||\pi_\beta) \leq \epsilon \qquad\qquad\qquad (4)$$

where $\mathcal{S}_{\text{safe}}$ represents the *largest safe region* as defined in Definition 3.2, which is the union of all safe regions and $\bar{D}$ is a normalized divergence metric considering the state in $\mathcal{S}_{\text{safe}}$ only (details in Appendix C). Here, we focus on minimizing the distance between the optimized policy and the behavior policy only for states within the safe region, while excluding behavior regularization for

unsafe states, which is also a key component for controlling behavior regularization. Any state that does not belong to this region indicates that no policy can satisfy the constraint. Therefore, the optimal solution for such states is to find the quickest escape sequence.

**Definition 3.2** *The largest safe region is defined as $\mathcal{S}_{safe} := \{s | \check{V}_c(s) \leq l\}$, and the smallest unsafe region is defined as $\mathcal{S}_{unsafe} := S \setminus \mathcal{S}_{safe}$.*

The value function $\check{V}_c(s)$ is the value function under the most **conservative** policy such that $\check{V}_c(s) = \min_\pi V_c^\pi(s)$. Our formulation (Eq. (4)) removes the constraint of controlling the distance from the behavior policy for states in the unsafe region, as it is unnecessary—no policy can satisfy the constraint in these states (as shown in Figure 1). Another advantage is that the largest safe region is easier to learn in the offline safe-RL setting (further discussion in the next section). A similar formulation can be found in FISOR (Zheng et al., 2024); however, our approach differs in several key aspects. We remove behavior cloning for states in the largest unsafe region, which is critical for learning a safe policy, especially when the behavior policy performs poorly in such states. Their feasible region is defined in terms of the value function for the hard constraint using Hamilton-Jacobi (HJ) reachability (Bansal et al., 2017) from control theory. However, this formulation applies only to deterministic systems (which is barely true in most of the RL environments) and cannot be generalized to problems with a more flexible cost limit $l$. Additionally, we differ significantly in the practical implementation of the algorithm, as detailed in Section 4.

In the following theorem, we demonstrate that the optimal solution of the decoupled optimization problem (4) is superior to the solution of the traditional optimization formulation (2):

**Theorem 3.3** *For any state $s \in \mathcal{S}_{safe}$ (defined in Definition 3.2), assuming $\pi_1^*$ is the optimal solution to the traditional optimization problem (2) and $\pi_2^*$ is the optimal solution to our decoupled behavior-aware region-based optimization problem (4), with $\rho = s$, then we have $V_r^{\pi_2^*}(s) \geq V_r^{\pi_1^*}(s)$, and $V_c^{\pi_2^*}(s) \leq l$. For any state $s \notin \mathcal{S}_{safe}$, with $\rho = s$, then the optimization traditional problem (2)* **does not** *have a solution, and $V_c^{\pi_2^*}(s) \leq V_c^\pi(s)$ for any other policy $\pi$.*

The reusts can also be easily generalized to the case when $\rho = \Delta(\mathcal{S}_{safe})$ and $\rho = \Delta(\mathcal{S}_{unsafe})$. The detailed proof is deferred to Section (C) in the Appendix due to page limit.

## 4 ALGORITHM

In this section, we formally introduce the details of our algorithm. Following the standard actor-critic approach, our algorithm includes five critic networks: $\hat{Q}_c$ and $\hat{V}_c$ to represent the cost $Q$-value function under the most **conservative** policy; $Q_r^\pi$ and $V_r^\pi$ to represent the reward $Q$-value function under the current policy $\pi$; and $Q_c^\pi$ to represent the cost $Q$-value function. We also use another actor neural network $z_\theta$ to represent the policy. Next, we introduce some key components of our algorithm (Alg. 1).

**Learning the least safe region:** Given the offline dataset $\mathcal{D}$, we will use the IQL(Kostrikov et al., 2022) to learn the value function $\check{V}_c(s)$, and the $Q-$ value function $\check{Q}_c(s, a)$, where $\check{V}_c(s) = \min_a \check{Q}_c(s, a)$ with expectile regression:

$$\mathcal{L}_{\check{V}_c} = \mathbb{E}_{(s,a)\sim\mathcal{D}} \left[ L^\tau(\check{Q}_c(s, a) - \check{V}_c(s)) \right], \tag{5}$$

$$\mathcal{L}_{\check{Q}_c} = \mathbb{E}_{(s,a,s')\sim\mathcal{D}} \left[ \left((1-\alpha)c(s,a) + \alpha(c(s,a) + \gamma \cdot \check{V}_c(s')) - \check{Q}_c(s,a)\right)^2 \right] \tag{6}$$

where $L^\tau(u) = |\tau - \mathbb{1}_{\{u>0\}}|u^2$, and $\mathbb{1}_{\{\cdot\}}$ is the indicator function. For $\tau \in (0.5, 1)$, this asymmetric loss places more weight when $\check{Q}_c \leq \check{V}_c$. Note that learning $\check{V}_c$ and $\check{Q}_c$ is independent of learning the agent, which means we can get an accurate estimation $\check{V}_c$ and $\check{Q}_c$ before training. Then the least("largest") safe region can be determined using $\check{V}_c$ and the cost limit $l$. Eq. (6) is referred to as the discounted safety Bellman equation (Fisac et al., 2019) and is used in practice (Zheng et al., 2024) to account for situations where there is a small probability, $1 - \alpha$, of transitioning to an absorbing state. The fixed point of the safety Bellman equation can be easily shown to converge to the fixed point of the Bellman equation as $\alpha \to 1$.

**Region-Based Risk-Reward Optimization:** As shown in the optimization (4), our practical algorithm also considers region-dependent objectives. For states in the safe region, where $\check{V}_c(s) \leq l$, we aim to maximize:

$$\frac{Q_r^\pi(s,a)}{Q_c^\pi(s,a)} \cdot \mathbb{1}_{\{Q_c^\pi(s,a) > \delta\}} + Q_r^\pi(s,a) \cdot \mathbb{1}_{\{Q_c^\pi(s,a) \leq \delta\}}, \tag{7}$$

where $a \sim \pi$ follows the actor network $z_\theta$. In the objective in Eq. (7), we penalize the reward $Q$-function based on the cost $Q$-value under the current policy, $Q_c^\pi(s,a)$, whenever $Q_c^\pi(s,a) > \delta$. The threshold $\delta$ is used to control potential unsafe actions, and to address estimation errors in critic networks. We simply set $\delta$ as the same as the cost limit $l$ in our simulation. In contrast, we maximize $Q_r^\pi(s,a)$ only when $Q_c^\pi(s,a) \leq \delta$, because the current policy is considered safe enough. The intuition behind this design is that the safe and unsafe regions are determined based on the value function under the most conservative policy. This implies that even starting from states in the safe region may not guarantee any violations under current policy $\pi$. Therefore, we need to be more cautious when optimizing the objective, a factor that is often overlooked in current literature.

For states in the unsafe region, we only need to train the policy to minimize the constraint $\check{Q}_c(s,a)$ as much as possible. Now we are ready to introduce our algorithm BARS (**B**ehavior-**A**ware **R**egion-based offline **S**afe RL):

$$\textbf{If } s \in \mathcal{S}_{\text{safe}} : \max_{\pi, a \sim \pi} \mathbb{E}\left[A_r^\pi(s,a)\right] \qquad\qquad \textbf{If } s \in \mathcal{S}_{\text{unsafe}} : \max_{\pi, a \sim \pi} \mathbb{E}\left[-\check{A}_c(s,a)\right]$$

$$\text{s.t.} \int_{\{a|\check{Q}_c(s,a) \leq l\}} \pi(a|s)da = 1, \forall s \in \mathcal{S}_{\text{safe}} \qquad \text{s.t.} \int_a \pi(\cdot|s)da = 1 \tag{8}$$

$$\bar{D}(\pi\|\pi_\beta) \leq \epsilon,$$

where $A_r^\pi(s,a) := \frac{Q_r^\pi(s,a)}{Q_c^\pi(s,a)} \cdot \mathbb{1}_{\{Q_c^\pi(s,a)>\delta\}} + Q_r^\pi(s,a) \cdot \mathbb{1}_{\{Q_c^\pi(s,a)\leq\delta\}}$, and $\check{A}_c(s,a) = \check{Q}_c(s,a) - \check{V}_c(s)$. BARS favors actions that provide cost-aware high rewards while ensuring the agent remains safe and maintains a small distributional shift from the behavior policy in the safe region. In the unsafe region, BARS focuses on finding the "least cost escape path" by minimizing the advantage function under the most conservative policy.

**Policy Update:** In BARS, we use a diffusion model $z_\theta$, as in (Zheng et al., 2024), to parameterize the policy network $\pi_\theta$. We follow a similar approach to (Zheng et al., 2024; Hansen-Estruch et al., 2023; Kang et al., 2024), considering a weighted loss to train the actor. However, our loss function is behavior-aware and region-based, incorporating the formulation in Eq. (8). Specifically, optimizing the policy in the safe region is achieved by minimizing the diffusion training loss for the behavior policy $\pi_\beta$: $\min_\theta \mathbb{E}_{t,(s,a),z}\left[\|z - z_\theta(a_t, s, t)\|_2^2\right]$ with the augmentation weight function $w_{\text{safe}}(s,a) = \exp(\mu_1 A_r^\pi(s,a) \cdot \mathbb{1}_{\{\check{Q}_c(s,a)\leq l\}})$ as:

$$\min_\theta \mathcal{L}_z^{\text{safe}}(\theta) := \min_\theta \mathbb{E}_{t \sim \mathcal{U}([0,T]), z \sim \mathcal{N}(0,I), (s,a) \sim D}\left[w_{\text{safe}}(s,a) \cdot \|z - z_\theta(a_t, s, t)\|_2^2\right], \tag{9}$$

where $a_t = \alpha_t a + \sigma_t z$ is the noisy action that satisfies the forward transition distribution $\mathcal{N}(a_t|\alpha_t a, \sigma_t^2 I)$ in the diffusion model, and $\alpha_t$ and $\sigma_t$ are human-designed noise schedules.

Similarly, optimizing the policy in the unsafe region without considering the distributional shift can be obtained by minimizing:

$$\min_\theta \mathcal{L}_z^{\text{unsafe}}(\theta) =: \min_\theta \mathbb{E}_{(s,a \sim \pi_\theta)}[\exp(-\mu_2 \check{A}_c(s,a))], \tag{10}$$

where $a \sim \pi_\theta$ can be obtained after obtaining $\theta$ by solving the diffusion ODEs/SDEs(Song et al., 2020).

**Remark 4.1** *To further improve the performance of BARS, we found it beneficial to incorporate an additional loss term, $D(\pi_t\|\pi_{t-k})$, into $\mathcal{L}_z^{unsafe}(\theta)$ at iteration $t$ after $K_{trust}$ training steps, where $k$ specifies how often we update the target policy, and we simply set $K_{trust}$ be the half of total training steps. This term helps ensure that the updated policy does not deviate significantly from the previous policy (as the trust region in TRPO (Schulman et al., 2015)) when a sufficiently good and safe policy has already been learned. Consequently, we propose a modified loss function for the unsafe region:*

$$\min_\theta \mathcal{L}_z^{unsafe}(\theta) =: \min_\theta \mathbb{E}_{(s,a \sim \pi_\theta)}[\exp(-\mu_2 \check{A}_c(s,a)) \cdot \|z_\theta(a_t, s, t) - z'(a_t, s, t)\|_2^2], \tag{11}$$

*where $z'$ is the copy of the policy network $\pi_{t-k}$. An ablation study comparing the results with and without the restrictions on the update can be found in Section 5.1. Note that $z'$ **does not represent the behavior policy** $\pi_\beta$. Thus, this regularization technique aligns with our motivation and remains consistent with the optimization problem in Eq.(4).*

Finally, combining the two loss functions above, we have the following weighted loss:

$$\min_\theta \mathcal{L}_z(\theta) := \min_\theta (b_1 \cdot \mathcal{L}_z^{\text{safe}}(\theta) + b_2 \cdot \mathcal{L}_z^{\text{unsafe}}(\theta)), \tag{12}$$

where $b_1 > 0, b_2 > 0$ are used to control the trade-off between two losses.

**Update reward/cost $Q$ functions under the current policy:** The cost and reward $Q-$value functions $Q_c^\pi$ and $Q_r^\pi$ are trained by minimizing the standard bellman loss using the offline dataset:

$$\mathcal{L}_{Q_r^\pi} = \mathbb{E}_{(s,a,s')\sim D, a'\sim\pi_\theta} \left[ (r(s,a) + \gamma \cdot Q_r^\pi(s',a') - Q_r^\pi(s,a))^2 \right] \tag{13}$$

$$\mathcal{L}_{Q_c^\pi} = \mathbb{E}_{(s,a,s')\sim D, a'\sim\pi_\theta} \left[ (c(s,a) + \gamma \cdot Q_c^\pi(s',a') - Q_c^\pi(s,a))^2 \right] \tag{14}$$

The full pseudo-code of BARS is presented in Algorithm 1.

---

**Algorithm 1 BARS** (Behavior-Aware Region-based offline Safe RL)

---

1: **Initialization:**
   offline dataset: $\mathcal{D}$, threshold: $\delta$, trust episode: $K_{\text{trust}}$, cost limit: $l$;
   initialize networks: $\check{V}_c, \check{Q}_c, Q_r, Q_c, z_\theta$, hyper-parameters: $\mu_1, \mu_2, b_1, b_2, k$;
2: **Determine the largest feasible region:**
3: **for** each gradient step **do**
4:     Update $\check{V}_c$ using Eq. (5).
5:     Update $\check{Q}_c$ using Eq. (6).
6: **end for**
7: Safe Region:   $\mathcal{S}_1 = \{s|\hat{V}_c(s) \leq l, \hat{Q}_c(s,a) \leq l\}$.
8: Unsafe Region:   $\mathcal{S}_2 = \{s|\hat{V}_c(s) > l\}$.
9: **for** each iteration **do**
10:     **Learning critic networks:**
11:         Update $Q_r$ using Eq. (13).
12:         Update $Q_c$ using Eq. (14).
13:     **Policy Update:**
14:         If $s \in \mathcal{S}_1$ then Calculate loss $\mathcal{L}_z^{\text{safe}}(\theta)$ using Eq. (9).
15:         If $s \in \mathcal{S}_2$ :
16:             If iteration $\leq K_{\text{trust}}$ then calculate loss $\mathcal{L}_z^{\text{unsafe}}(\theta)$ using Eq. (10).
17:             If iteration $> K_{\text{trust}}$ then calculate loss $\mathcal{L}_z^{\text{unsafe}}(\theta)$ using Eq. (11).
18:         Calculate the policy loss for the offline dataset $\mathcal{D}$ using Eq. (12).
19:         Update policy $z_\theta$. Set $z' \leftarrow z_\theta$ every $k$ episode.
20: **end for**

---

## 5 Experiments

**Evaluation Setups.** We conduct extensive evaluations using Safety-Gymnasium tasks from the DSRL (Liu et al., 2023a) benchmark, enabling us to assess the performance of our algorithm compared to state-of-the-art offline safe reinforcement learning methods. Additionally, to better illustrate the advantage of BARS under different data quality conditions which has never been investigated in safe offline RL literature, we compare BARS with other baselines in three distinct scenarios: (1) high-quality offline dataset ( predominately safe data), (2) medium-quality offline dataset (balanced safe and unsafe data), and (3) low-quality offline dataset (predominately unsafe data). To implement these scenarios, we create three data distributions by selectively sampling the original safe data and duplicating the unsafe data, resulting in safe-to-unsafe data ratios of $8 : 1, 1 : 1$, and $1 : 3$, respectively. We employ *normalized return* and *normalized cost* as evaluation metrics; further details are provided in Appendix D.4. Notably, a normalized cost below 1 signifies safety. Following the principles of DSRL, safety is prioritized as the primary criterion for evaluation. The objective is to

Table 1: Normalized DSRL benchmark results. ↑ means the higher the better. ↓ means the lower the better. Each value is averaged over 20 evaluation episodes. Gray: Unsafe agents. **Bold**: Safe agents whose normalized cost is smaller than 1. **Blue**: Safe agents with the highest reward.

| Task | BC-safe | | CPQ | | COptiDICE | | FISOR | | **BARS**(ours) | |
|---|---|---|---|---|---|---|---|---|---|---|
| | reward ↑ | cost ↓ | reward ↑ | cost ↓ | reward ↑ | cost ↓ | reward ↑ | cost ↓ | reward ↑ | cost ↓ |
| CarButton1 | **0.006** | **0.49** | 0.49 | 26.65 | -0.09 | 4.41 | **-0.11** | **0.73** | **-0.04** | **0.51** |
| CarButton2 | -0.03 | 1.1 | 0.20 | 32.59 | -0.02 | 3.76 | **-0.001** | **0.83** | **-0.04** | **0.27** |
| CarGoal1 | **0.15** | **0.82** | 0.60 | 4.87 | 0.47 | 1.68 | 0.42 | 1.63 | **0.38** | **0.48** |
| CarGoal2 | 0.10 | 3.67 | 0.18 | 4.09 | 0.12 | 1.76 | **0.03** | **0.24** | **0.07** | **0.44** |
| CarPush1 | **0.18** | **0.82** | 0.10 | 5.21 | 0.23 | 1.54 | 0.24 | 1.61 | **0.18** | **0.47** |
| CarPush2 | 0.08 | 1.43 | 0.07 | 13.76 | 0.16 | 9.38 | 0.10 | 1.21 | **0.148** | **0.61** |
| AntVelocity | **0.88** | **0.4** | **-1.01** | **0** | 0.99 | 11.75 | **0.89** | **0.003** | **0.928** | **0.211** |
| HalfCheetah | **0.83** | **0.04** | 0.675 | 14.85 | **0.61** | **0** | **0.89** | **0** | **0.966** | **0.08** |
| SwimmerVel | **0.48** | **0.7** | **0.001** | **0** | 0.7 | 36.3 | **-0.04** | **0.31** | -0.05 | 0.125 |
| Average | 0.297 | 1.052 | 0.145 | 11.3 | 0.35 | 7.84 | **0.268** | **0.72** | **0.282** | **0.35** |

achieve higher rewards while ensuring strict adherence to safety standards. To accurately simulate safety-critical scenarios, we impose stricter safety regulations. Specifically, the cost limit for all the Safety-Gymnasium tasks is set as 10.

**Baselines.** We compare our algorithm with the following baselines: CPQ(Xu et al., 2022): Designate actions outside the data distribution as unsafe and update the reward critic with safe state-action pairs; COptiDICE(Lee et al. (2022)): Estimates the stationary distribution corrections for the optimal policy in terms of returns with an upper bound on costs; FISOR(Zheng et al. (2024)): Learn the feasible region to maximize rewards within it while minimizing constraint violations in the unfeasible region.

**Main Results on the original dataset.** The experimental results on the original dataset (Liu et al., 2023a) are shown in Table 1. Our algorithm consistently achieves a safe policy across all tasks and obtains the highest reward in most environments. While FISOR[1] also ensures safety across tasks and achieves higher rewards in some cases; CPQ and COptiDICE fail to learn a safe policy in almost all tasks. We can observe that our algorithm is comparable to the state-of-the-art algorithm.

**Main Results Across Datasets of Varying Qualities.** As shown in Table 2, under different dataset distributions, none of the existing state-of-the-art methods can consistently ensure a safe policy as the data ratio varies, even slightly. Our algorithm is the **only one** capable of maintaining a **safe** policy across all tasks, regardless of data distribution variations, while also achieving the highest overall rewards in every data ratio scenario. This highlights the robustness and adaptability of our approach, demonstrating its effectiveness in environments where the proportion of safe to unsafe data fluctuates significantly. These results also validate our findings regarding the existing challenges in offline safe RL and emphasize the importance of a behavior-aware, region-based policy.

### 5.1 ABLATION STUDY

**Different Cost Limits.** To demonstrate the ability of BARS to perform well even under different cost limits, we evaluate BARS on CarGoal2 with cost limits of 10, 20, and 40. As shown in Table 3, BARS is able to achieve higher rewards as the cost limit increases. BARS proves more adaptable to practical situations by adjusting the cost limit, as a higher limit allows the agent greater freedom to explore higher-reward policies.

**Safe offline RL with new Data Collection.** To further investigate the issue of data quality in safe offline RL and verify BARS's ability to train a behavior-aware, region-based policy, we design a new experimental setting where the agent is allowed to collect a *new dataset* during training. This setting has significant practical value, as it reflects a scenario where a policy is initially trained on a given dataset (maybe not good) but can further improve its performance by collecting additional data when the policy is deemed **safe** enough for exploration. In this experiment, we first train BARS for $2e5$ steps using a dataset randomly selected from the original offline dataset (with a safe-to-

---

[1]The results are obtained using the official code published by the FISOR authors: https://github.com/ZhengYinan-AIR/FISOR. Unfortunately, we were unable to reproduce the exact results reported in the original paper for some environments. However, our results are still comparable with those reported $(0.30, 0.35)$.

Table 2: Normalized DSRL benchmark results with Data Vaires. ↑ means the higher the better. ↓ means the lower the better. Each value is averaged over 20 evaluation episodes. Gray: Unsafe agents. **Bold**: Safe agents whose normalized cost is smaller than 1. **Blue**: Safe agents with the highest reward.

| Task | Data Ratio | | CPQ | | COptiDICE | | FISOR | | **BARS**(ours) | |
|------|------|------|------|------|------|------|------|------|------|------|
| | Safe | Unsafe | reward ↑ | cost ↓ | reward ↑ | cost ↓ | reward ↑ | cost ↓ | reward ↑ | cost ↓ |
| CarButton1 | 8 | 1 | 0.25 | 46.3 | 0.04 | 2.7 | 0.01 | 1.12 | **-0.15** | **0.535** |
| | 1 | 1 | 0.03 | 22.3 | **-0.09** | **0.86** | 0.01 | 1.045 | **-0.07** | **0.565** |
| | 1 | 3 | 0.10 | 44.4 | -0.23 | 1.38 | -0.09 | 1.455 | **-0.25** | **0.1** |
| CarButton2 | 8 | 1 | 0.21 | 50.08 | **-0.02** | **0.93** | 0.0083 | 1.23 | **-0.10** | **0.34** |
| | 1 | 1 | 0.04 | 39.40 | -0.12 | 3.65 | -0.02 | 1.75 | **-0.08** | **0.63** |
| | 1 | 3 | 0.11 | 27.01 | -0.29 | 3.54 | -0.006 | 2.655 | **-0.25** | **0.56** |
| CarGoal1 | 8 | 1 | 0.21 | 3.05 | 0.36 | 3 | 0.458 | 1.505 | **0.35** | **0.52** |
| | 1 | 1 | 0.25 | 5.92 | 0.26 | 1.48 | **0.165** | **0.75** | **0.21** | **0.16** |
| | 1 | 3 | 0.59 | 3.995 | 0.22 | 2.01 | 0.183 | 1.505 | **0.08** | **0.21** |
| CarGoal2 | 8 | 1 | 0.001 | 6.84 | 0.24 | 3.23 | **0.03** | **0.355** | **0.03** | **0.505** |
| | 1 | 1 | 0.004 | 4.97 | 0.14 | 2.16 | 0.06 | 2.49 | **0.002** | **0.71** |
| | 1 | 3 | 0.24 | 20.07 | 0.08 | 1.97 | 0.14 | 2.08 | **0.001** | **0.685** |
| CarPush1 | 8 | 1 | 0.28 | 1.21 | 0.23 | 2.165 | **0.27** | **0.54** | **0.18** | **0.53** |
| | 1 | 1 | **-0.02** | **0.085** | 0.22 | 1.26 | 0.27 | 2.12 | **0.11** | **0.54** |
| | 1 | 3 | -0.01 | 7.49 | 0.25 | 3.60 | 0.20 | 1.67 | **0.10** | **0.43** |
| CarPush2 | 8 | 1 | -0.04 | 17.395 | 0.05 | 2.42 | **0.07** | **0.24** | **0.12** | **0.56** |
| | 1 | 1 | 0.04 | 9.95 | 0.11 | 4.91 | 0.08 | 1.82 | **0.01** | **0.41** |
| | 1 | 3 | 0.04 | 34.88 | 0.15 | 4.99 | -0.01 | 1.355 | **0.004** | **0.49** |
| AntVelocity | 8 | 1 | **-1.012** | **0** | 0.96 | 3.285 | **0.88** | **0** | **0.96** | **0.32** |
| | 1 | 1 | **-1.011** | **0** | 0.99 | 13.99 | **0.78** | **0** | **0.81** | **0.045** |
| | 1 | 3 | **-1.011** | **0** | 0.97 | 11.62 | **0.77** | **0** | 0.71 | 0.047 |
| HalfCheetah | 8 | 1 | **0.019** | **0.215** | 0.51 | 0 | 0.87 | 0 | **0.88** | **0.005** |
| | 1 | 1 | **-0.19** | **0** | 0.50 | 0 | 0.83 | 0 | **0.88** | **0** |
| | 1 | 3 | **-0.25** | **0** | 0.46 | 0 | 0.84 | 0 | **0.86** | **0.005** |
| SwimmerVel | 8 | 1 | **0.002** | **0** | 0.48 | 5.4 | **-0.06** | **0.125** | **-0.05** | **0.86** |
| | 1 | 1 | 0.1 | 1.35 | 0.50 | 15.1 | 0.312 | 1.24 | **-0.03** | **0.54** |
| | 1 | 3 | 0.03 | 1.02 | -0.05 | 1.465 | 0.33 | 2.185 | **-0.04** | **0.705** |
| Average | — | — | -0.03 | 12.88 | 0.26 | 3.59 | 0.27 | 1.08 | **0.195** | **0.407** |

Table 3: Various cost limit for CarGoal2 task. Each value is averaged over 20 evaluation episodes under cost limit 10, 20 and 40.

| Methods | cost limit 10 | | cost limit 20 | | cost limit 40 | |
|---------|------|------|------|------|------|------|
| | reward ↑ | cost ↓ | reward ↑ | cost ↓ | reward ↑ | cost ↓ |
| BARS | 0.07 | 0.44 | 0.10 | 0.24 | 0.20 | 0.75 |

unsafe ratio of $1:1$), We then use the trained policy to collect a new dataset(First-Collected) and continue training the policy for another $2e5$ steps, then repeat this process to collect another new dataset(Second-Collected) and train for an additional $2e5$ steps to assess performance improvement. It is clear to observe that the mean episode cost drops dramatically (from $69.72$ to $7.45$) even after one new collection(Figure 3(c)), which implies that the policy is already **safe enough** for further exploration. Besides, we report the performance of BARS w/wo new data collection after training for $4e5, 6e5$ steps in Figure 3(a,b). We can observe that allowing new data collection during learning can gradually further improve performance. The reward is increased by **72**% after training for $4e5$ steps and is increased by **26**% after training for $6e5$ steps while all the policies are ensured to be safe. Detailed experiment settings are available in Section E in the Appendix.

**Adding Restrictions when Updating the Policy for States in the Unsafe Region.** As discussed in Section 1, adding the restriction when updating the policy for states in the unsafe region can restrict the policy from diverging from too much. In this section, we evaluate BARS's performance by conducting experiments both with and without the additional loss term detailed in Eq. (11). The experimental results in Table 4 demonstrate that adding the policy restriction term can help boost the reward. Note that without adding this regularization, BARS can still ensure a safe policy.

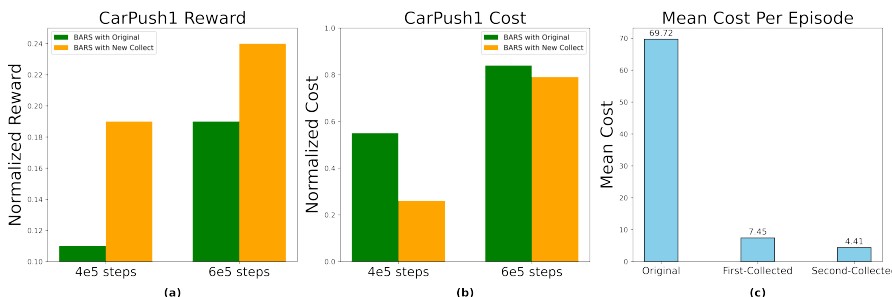

Figure 3: (a), (b): Normalized reward and cost during training on CarPush 1; (c): The average cost per episode (cost limit $l = 10$): the original dataset has a safe-to-unsafe ratio of $1 : 1$. The first-collected dataset is generated using the policy after training for $2e5$ steps, while the second-collected dataset is generated using the policy after training for $4e5$ steps.

Table 4: Restriction Update Result

| Task | w/o policy restriction | | with policy restriction | |
|---|---|---|---|---|
| | reward ↑ | cost ↓ | reward ↑ | cost ↓ |
| SwimmerVel | -0.06 | 0.02 | -0.05 | 0.125 |
| AntVel | 0.75 | 0.01 | 0.928 | 0.211 |
| HalfCheetah | 0.36 | 0 | 0.966 | 0.08 |

## 6 RELATED WORK

In offline safe RL, primal-dual-based approaches are the most commonly used (Tessler et al., 2018; Chow et al., 2017; Ding et al., 2020; Efroni et al., 2020; Wei et al., 2022a; Fujimoto et al., 2019; Kumar et al., 2019). CPQ (Xu et al., 2022) employs a VAE to detect and penalize out-of-distribution (OOD) actions based on their associated costs. COptiDICE (Lee et al., 2022), an extension of OptiDICE (Lee et al., 2021), integrates safety constraints and derives a safe policy from the stationary distribution of the optimal policy. Other methods, such as applying decision transformers (Chen et al., 2021) or diffuser models (Janner et al., 2022), have been used to study safe offline RL, though they tend to be computationally inefficient. FISOR (Zheng et al., 2024) decouples the process of satisfying safety constraints from reward maximization, leveraging a diffusion model for policy generation. VOCE (Guan et al., 2024) addresses extrapolation errors caused by OOD actions by estimating both cost and reward Q-values in a pessimistic manner. Additionally, some works (Yang et al., 2023; Wabersich et al., 2023; Kim et al., 2024) draw from control theory to enforce state-wise constraints. More discussions on the related work can be found in Section A in the Appendix.

## 7 CONCLUSION

We propose BARS, a novel algorithm that tackles the existing issues of applying behavior regularization in offline safe RL. We demonstrate that directly applying it in offline safe RL often leads to issues such as policy infeasibility and the risk of imitating poor or unsafe behavior. BARS addresses these challenges by distinguishing between safe and unsafe states and applying region-specific, selective behavior regularization to optimize the policy accordingly. Extensive experiments demonstrate that BARS significantly outperforms several state-of-the-art baselines in both rewards and safety, particularly when the behavior policy is far from optimal. Even in scenarios where the dataset quality is low, BARS continues to perform robustly, while other baselines fail to maintain safe policies across environments. This highlights BARS's potential to resolve a previously overlooked issue in offline safe RL and provide a framework for safer and more effective policy learning.

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

# A  MORE RELATED WORKS

Safe RL has attracted great attention recently, both in online setting (Efroni et al., 2020; Liu et al., 2021; Bura et al., 2022; Wei et al., 2022b;a; 2024a; Müller et al., 2023; Germano et al., 2023; Ding et al., 2024; Müller et al., 2024) and offline setting (Wu et al., 2021; Chen et al., 2021; Lee et al., 2022; Liu et al., 2023b; Xu et al., 2022; Guan et al., 2024; Zheng et al., 2024; Hong and Tewari, 2024).

In theoretical offline RL, certain assumptions about the ratio between the state-action occupancy distribution of a policy $\pi$ and the dataset distribution $\mu$ are necessary to prove convergence. This ratio is represented by $w^\pi = d^\pi/\mu$. A commonly used assumption is that the $\ell_\infty$ concentrability $C_{\ell_\infty}^\pi$, defined as the infinite norm of $w^\pi$, is bounded for all policies (Liu et al., 2019; Chen and Jiang, 2019; Wang et al., 2019; Liao et al., 2022; Zhang et al., 2020). However, this assumption is difficult to satisfy, particularly in offline safe RL, as it requires the dataset to adequately cover all unsafe state-action pairs. To address this, recent works (Rashidinejad et al., 2021; Zhan et al., 2022; Chen and Jiang, 2022; Xie et al., 2021; Uehara and Sun, 2021) introduce pessimism to reduce the requirement to only the best-covered policy. Zhu et al. (2023) further relaxes this assumption by introducing $\ell_2$ concentrability, which is always bounded by $\ell_\infty$. While existing offline safe RL studies (Hong et al., 2024; Le et al., 2019) still require coverage for all policies, a recent work Wei et al. (2024b) relaxes this assumption to single-policy coverage.

Deep offline safe RL algorithms (Lee et al., 2022; Liu et al., 2023b; Xu et al., 2022; Chen et al., 2021; Zheng et al., 2024) have shown strong empirical performance but usually lack theoretical guarantees. As we mentioned in the introduction, balancing generalization with preventing undesirable behaviors from out-of-distribution actions is a key challenge in offline RL. A straightforward approach is to directly constrain the learned policy to align with the behavior policy used to collect the data set (Fujimoto et al., 2019; Kumar et al., 2019; Fujimoto and Gu, 2021; Wu et al., 2019). Other methods address this by making conservative estimates of future rewards through a lower bound on the true value function (Kumar et al., 2020; Yu et al., 2021) or by penalizing out-of-distribution (OOD) actions to control distributional shift (Kostrikov et al., 2021; Lyu et al., 2022). Model-based approaches often handle uncertainty by using ensembles, penalizing actions with high variability across models and promoting those that show consistency (Janner et al., 2019; Kidambi et al., 2020). Recently, some studies have explored the use of expressive diffusion models for policy learning to capture more complex distributions, achieving impressive results (Janner et al., 2022; Lu et al., 2023; Hansen-Estruch et al., 2023; Zheng et al., 2024).

# B  MORE DISCUSSIONS ON THE COMPARISON BETWEEN BARS AND FISOR

Our formulation is inspired by FISOR (Zheng et al., 2024), which proposes a feasibility-guided diffusion model for offline safe RL. However, our formulation differs from FISOR in several significant ways:

- To the best of our knowledge, this is the first paper to address the issue that using behavior regularization constraints in safe RL can lead to infeasible solutions, particularly when multiple constraints are involved. Moreover, adding behavior regularization may result in the learned policy imitating the behavior policy, even in states where the behavior policy performs poorly or is unsafe. This issue is not limited to FISOR; almost all existing algorithms fail to account for this problem and cannot reliably guarantee a safe policy when the behavior policy is suboptimal, as demonstrated in Table 2.

- FISOR addresses only the "hard constraint" setting, where the cost value function represents the maximum constraint violation (largest state-wise cost) in the trajectory. This greatly limits the generalization of their approach to other scenarios where a cost limit is considered. Since ristrictly control the hard constraint on some dangerous states can also be done by adding a large penalty to the reward function for such states to avoid a unsafe policy. In contrast, BARS can handle a wide range of constraints, offering flexibility by adjusting the constraint limit in the algorithm.

- FISOR (Zheng et al., 2024) defines the feasible region using the value function for the hard constraint based on Hamilton-Jacobi (HJ) reachability (Bansal et al., 2017) from control theory. However, the formulation in FISOR applies only to **deterministic systems**, which is rarely the case

in most RL environments, and cannot be generalized to problems with more flexible cost limits. The Bellman equation used in FISOR to train the value function no longer holds in stochastic systems.

- FISOR optimizes the policy by maximizing the value function to achieve the highest reward without considering constraints, which overlooks the fact that such a policy may not be safe. In contrast, we propose a "Region-Based Risk-Reward Optimization" in Eq.(7), which accounts for potential errors in value function estimation (likely to be true in practice) and directly optimizes the policy based on the **current** policy. This approach enables BARS to avoid potential unsafe actions effectively.

## C   PROOF OF THEOREM 3.3

For any state $s \in \mathcal{S}$, define the standard optimization problem as:

$$\max_{\pi} V_r^{\pi}(s) \quad \text{s.t.} \quad V_c^{\pi}(s) \leq l, \ D(\pi||\pi_{\beta}) \leq \epsilon, \tag{15}$$

Our decoupled behavior-aware, region-based formulation is:

$$\text{If} \quad s \in \mathcal{S}_{\text{safe}} : \max_{\pi} V_r^{\pi}(s); \qquad\qquad \text{If} \quad s \in \mathcal{S}_{\text{unsafe}} : \max_{\pi}(-V_c^{\pi}(s))$$

$$\text{s.t.} \quad V_c^{\pi}(s) \leq l, \bar{D}(\pi||\pi_{\beta}) \leq \epsilon, \tag{16}$$

where we consider the KL divergence as the normalized distance metric $\bar{D}$, which is defined as: $\bar{D} := \sum_{s \in \mathcal{S}_{\text{safe}}} \int_a \pi(a|s) \log\left(\frac{\pi(a|s)}{\pi_{\beta}(a|s)}\right) da$. Here, we focus on minimizing the distance between the optimized policy and the behavior policy only for states within the safe region, while excluding behavior regularization for unsafe states. Let $\pi_1^*$ be the optional solution of the optimization problem (15), and $\pi_2^*$ be the optimal solution of the decoupled optimization problem (16). To show that $\pi_2^*$ dominates $\pi_1^*$, we will show that $\pi_1^*$ is a feasible solution to the optimization problem (16). First, for any $s \in \mathcal{S}_{\text{safe}}$, we have that $V_c^{\pi_1^*}(s) \leq l, D(\pi_1^*||\pi_{\beta}) \leq \epsilon$ according to the constraints in (15). Since we have

$$\epsilon \geq D(\pi_1^*||\pi_{\beta}) = \sum_{s \in \mathcal{S}} \int_a \pi_1^*(a|s) \log\left(\frac{\pi_1^*(a|s)}{\pi_{\beta}(a|s)}\right) da$$

$$= \sum_{s \in \mathcal{S}_{\text{safe}}} \int_a \pi_1^*(a|s) \log\left(\frac{\pi_1^*(a|s)}{\pi_{\beta}(a|s)}\right) da + \sum_{s \in \mathcal{S}_{\text{unsafe}}} \int_a \pi_1^*(a|s) \log\left(\frac{\pi_1^*(a|s)}{\pi_{\beta}(a|s)}\right) da$$

$$\geq \sum_{s \in \mathcal{S}_{\text{safe}}} \int_a \pi_1^*(a|s) \log\left(\frac{\pi_1^*(a|s)}{\pi_{\beta}(a|s)}\right) da.$$

Thus, $\pi_1^*$ is also a feasible solution to the optimization problem (16) as well. Therefore we have $V_r^{\pi_2^*}(s) \geq V_r^{\pi_1^*}(s)$. For the case when $s \notin \mathcal{S}_{\text{safe}}$, the optimization problem (15) **does not** have a solution. $V_c^{\pi_2^*}$ is the **smallest** cumulative cost we can have.

## D   EXPERIMENTAL DETAILS

### D.1   REACH-AVOID TASK

The experimental setup illustrated in Figure 1, as adapted from the toy case experiment discussed in (Zheng et al., 2024), entails an agent tasked with navigating towards a goal while circumventing a hazard located at the center of the map. The state space of the agent is defined as $S := (x, y, v, \theta)$, where $x$ and $y$ are the agent's coordinates, $v$ represents the initial velocity, and $\theta$ the direction of movement. The action space is designated as $A := (\bar{v}, \bar{\theta})$, with $\bar{v}$ denoting acceleration and $\bar{\theta}$ angular acceleration. The reward function $r$ evaluates performance by measuring the change in distance to the target between consecutive time steps. To encourage the agent to reach the goal location, we assign a substantial reward when the agent successfully arrives at the goal. The cost function $h$ is expressed as:

$$h := R_{\text{hazard}} - d_{\text{hazard}}$$

In this expression, $R_{\text{hazard}}$ represents the radius of the hazard, set to 0.8 in this implementation, and $d_{\text{hazard}}$ denotes the distance between the agent and the hazard. A negative or zero value of $h$ indicates a collision with the hazard, signaling a breach of safety. The compliance measure $c$ is then defined as $c := \mathbb{1}_{\{h \leq 0\}}$, which assesses the agent's adherence to safety protocols during navigation.

To gather sufficient data, we first allow the agent to interact with the showcase environment and train it using the TD3-Lag algorithm, as modified by (Fujimoto et al., 2018). Once the agent is well-trained, it is deployed to collect expert data. Additionally, we run a random policy to accumulate a large number of unsafe or low-quality data samples. These random policy samples are then combined with the expert data to generate three distinct categories of offline datasets: high-quality, with a safe-to-unsafe ratio of 19:1; medium-quality, with a ratio of 10:1; and low-quality, with a ratio of 5:1. The distribution of the collected data is illustrated in Figure 4, where the regions with higher data concentration, particularly around hazard zones for the low-quality dataset, are highlighted in bright colors.

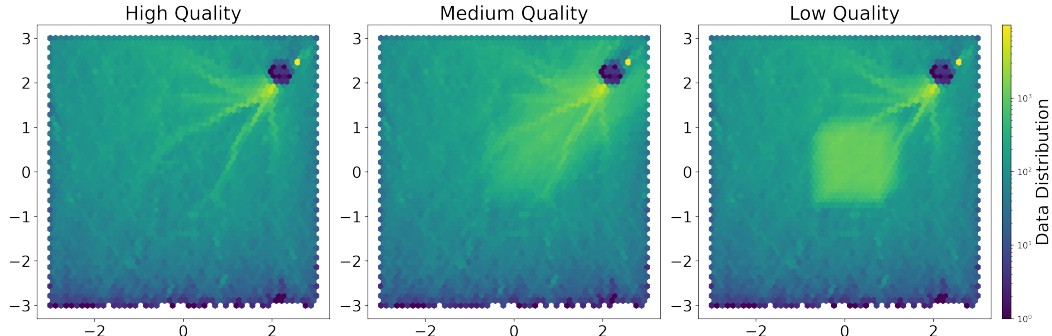

Figure 4: Data Distribution for High/Medium/Low Quality Offline Dataset

To more effectively demonstrate BARS's advantages across varying data quality scenarios (results in Table 2), we compared BARS with other baseline methods on Safety-Gymnasium tasks under three distinct data conditions: (1) a high-quality offline dataset predominantly composed of safe data, (2) a medium-quality offline dataset with a balanced mix of safe and unsafe data, and (3) a low-quality offline dataset largely consisting of unsafe data. To establish these scenarios, we created three different data distributions by selectively sampling the existing safe data and augmenting the unsafe data at the beginning of training. This approach resulted in safe-to-unsafe data ratios of 8:1, 1:1, and 1:3, respectively, with the exact data quantities shown in Table 5.

## D.2 STARTING FROM UNSAFE REGION IN THE TOY EXAMPLE

Figure 5 illustrates that if the agent starts in a hazardous region (unsafe region), BARS can still learn a reasonable policy that reaches the destination while escaping the hazardous region as quickly as possible with a lower cost than FISOR.

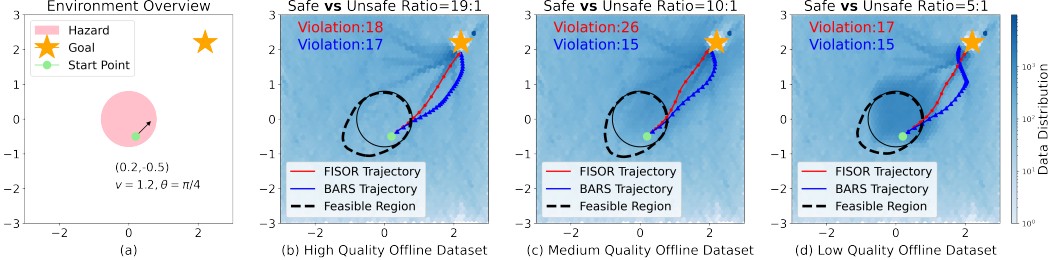

Figure 5: Starting at Unsafe Region

Table 5: Summary of Simulation Data Samples

| Task | Data Ratio | | Dataset | |
|------|------|------|------|------|
| | Safe | Unsafe | Number of Safe Samples | Number of Unsafe Samples |
| CarButton1 | 8 | 1 | 1,600,000 | 200,000 |
| | 1 | 1 | 1,600,000 | 1,600,000 |
| | 1 | 3 | 1,600,000 | 4,800,000 |
| CarButton2 | 8 | 1 | 1,600,000 | 200,000 |
| | 1 | 1 | 1,600,000 | 1,600,000 |
| | 1 | 3 | 1,600,000 | 4,800,000 |
| CarGoal1 | 8 | 1 | 560,000 | 70,000 |
| | 1 | 1 | 560,000 | 560,000 |
| | 1 | 3 | 560,000 | 1,680,000 |
| CarGoal2 | 8 | 1 | 1,600,000 | 200,000 |
| | 1 | 1 | 1,600,000 | 1,600,000 |
| | 1 | 3 | 1,600,000 | 4,800,000 |
| CarPush1 | 8 | 1 | 1,200,000 | 150,000 |
| | 1 | 1 | 1,200,000 | 1,200,000 |
| | 1 | 3 | 1,200,000 | 3,600,000 |
| CarPush2 | 8 | 1 | 1,600,000 | 200,000 |
| | 1 | 1 | 1,600,000 | 1,600,000 |
| | 1 | 3 | 1,600,000 | 4,800,000 |
| AntVelocity | 8 | 1 | 1,600,000 | 200,000 |
| | 1 | 1 | 1,600,000 | 1,600,000 |
| | 1 | 3 | 1,600,000 | 4,800,000 |
| HalfCheetah | 8 | 1 | 1,600,000 | 200,000 |
| | 1 | 1 | 1,600,000 | 1,600,000 |
| | 1 | 3 | 1,600,000 | 4,800,000 |
| SwimmerVel | 8 | 1 | 1,200,000 | 150,000 |
| | 1 | 1 | 1,200,000 | 1,200,000 |
| | 1 | 3 | 1,200,000 | 3,600,000 |

### D.3 SAFETY-GYMNASIUM

Safety-Gymnasium (Ray et al., 2019; Ji et al., 2023) is a highly modular and easily customizable benchmark environment library, built on MuJoCo, and designed to support research in Safe RL. The four distinct agents are "Car," "Ant," "HalfCheetah," and "Swimmer," while the tasks associated with these agents are "Button," "Goal," "Push," and "Vel." The numbers "1" and "2" indicate the difficulty levels of these tasks. Figure 6 provides visualizations of all the tasks.

### D.4 EVALUATION METRICS

We employ normalized reward and cost as comparison metrics as proposed by (Liu et al., 2023a). Let $r_{\max}(M)$ and $r_{\min}(M)$ represent the maximum and minimum empirical reward returns for task M; detail values can be found in Table 6, respectively. The normalized reward is calculated as follows:

$$R_{\text{normalized}} = \frac{R_\pi - r_{\min}(M)}{r_{\max}(M) - r_{\min}(M)},$$

where $R_\pi$ indicates the evaluated reward return of policy $\pi$. In contrast, normalized cost is defined using a different formula to enhance result differentiation. It is computed based on the ratio between the evaluated cost return $C_\pi$ and the cost limit $l$ :

$$C_{\text{normalized}} = \frac{C_\pi + \epsilon}{l + \epsilon},$$

where $\epsilon$ is a positive number to ensure numerical stability if the cost limit $l$ is 0. Note that the cost return and the cost limit are always non-negative. Therefore, both $C_{\text{normalized}}$ and $R_{\text{normalized}}$ fall within the range [0,1], where a lower cost is preferable and a higher reward is more desirable.

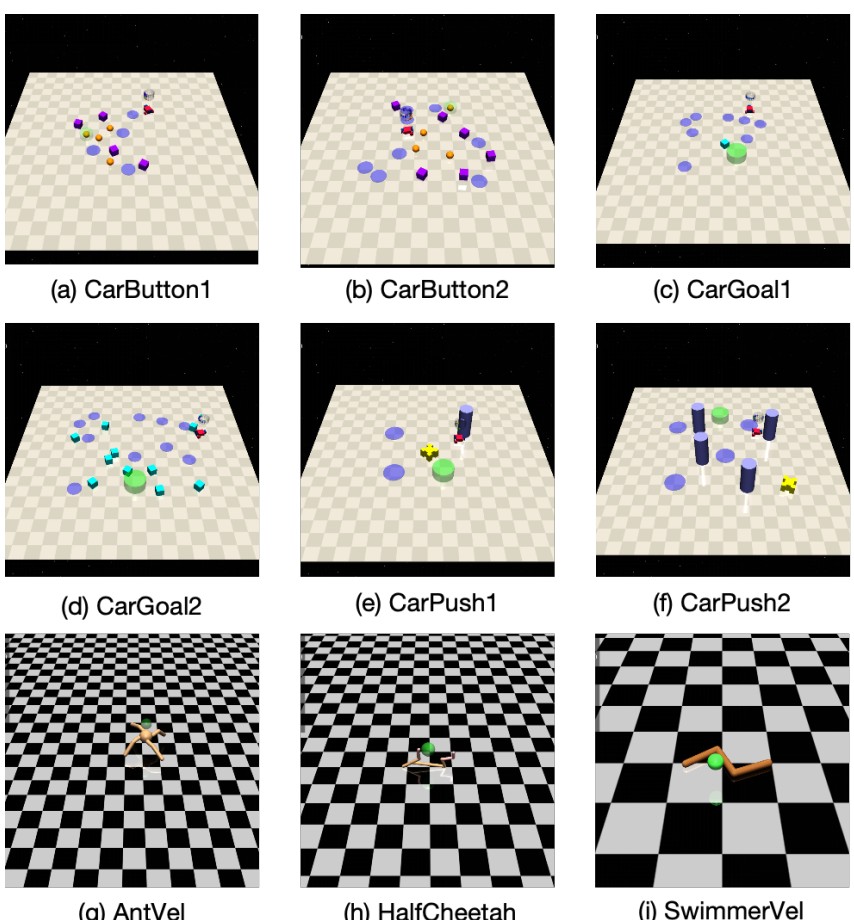

Figure 6: Safety-Gymnasium Visualization Overview

Table 6: Exact value for $r_{\text{max}}$ and $r_{\text{min}}$

| Task | $r_{\text{max}}$ | $r_{\text{min}}$ |
|------|------|------|
| CarButton1 | 44.42 | 0.0028 |
| CarButton2 | 41.99 | 0.0017 |
| CarGoal1 | 39.90 | 0.012 |
| CarGoal2 | 28.90 | 0.001 |
| CarPush1 | 16.30 | 0.013 |
| CarPush2 | 15.14 | 0.0002 |
| AntVel | 2976.27 | 6.15 |
| HalfCheetah | 2806.93 | 5.75 |
| SwimmerVel | 238.95 | 0.071 |

## D.5 HYPERPARAMETERS

The entire training process for each task was conducted on a single RTX-4090 GPU, taking approximately two hours per task. All hyperparameters used in the experiments are detailed in Table 7.

**Training Details** We structured our training process into a two-stage approach to enhance efficiency. In the first stage, we train and save the critical components, $\tilde{Q}_c$ and $\tilde{V}_c$. In the second stage, we load these pre-trained components and proceed to update $Q_r$, $Q_c$, and $z_\theta$. This bifurcated training strategy significantly reduces the overall time required for model convergence. Throughout the training process, we employ the Adam optimizer and opt not to use weight decay regularization to

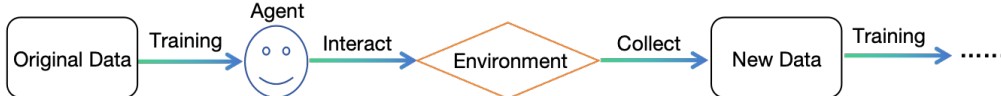

Figure 7: Training Process with New Dataset Collection

avoid potential negative impacts on training dynamics. Different tasks require different parameters. For the loss weight controller $b_1, b_2$, in simpler task environments ("AntVel" and "HalfCheetah") where learning a safe policy is easier, we prioritize maximizing rewards by setting $b_1 = 3, b_2 = 1$. Conversely, in more complex environments where learning a safe policy is challenging, we adjust the parameters to $b_1 = 1, b_2 = 3$ to emphasize safety. Additionally, we introduce an extra loss term as detailed in Eq. (11), and update the reference network $z'$ every $k = 5$ steps.

Table 7: Hyperparameters of BARS

| Parameter | Value |
|---|---|
| Activation function | ReLU |
| Expectile $\tau$ | 0.9 |
| Discount factor $\gamma$ | 0.99 |
| Soft update factor | 0.001 |
| $\alpha$ | 0.99 |
| Temperature $\mu_1$ | 3 |
| Temperature $\mu_2$ | 5 |
| Safe Loss Control $b_1$ | 1 |
| Unsafe Loss Control $b_2$ | 3 |
| Number of times Gaussian noise is added $T$ | 5 |
| Number of action candidates $N$ | 16 |
| Diffusion Model | DDPM (Hansen-Estruch et al., 2023) |
| $K_{\text{trust}}$ | 5e5 |
| Training steps | 1e6 |
| $k$ | 5 |

## E    ABLATION STUDIES

### E.1    COLLECT NEW DATA TO IMPROVE POLICY GRADUALLY

In this section, we outline the methodology for collecting new data using a pre-trained policy. Initially, we use the dataset from (Liu et al., 2023a) to randomly select a mini dataset for training our first agent, $\pi_1$, for $2e5$ steps as a safe policy. Following this, we transition from an offline setting to an online environment for collecting new data. In this online mode, the agent interacts directly with the environment, executing the learned policy while observing rewards and any violations of predefined costs. These new data samples are then used to further train the policy for another $2e5$ steps. We repeat this process for additional training rounds. In the experiment, we select a dataset for CarPush1, which includes 600,000 data samples, distributed across 600 trajectories, each containing 1,000 steps. Each data collection process is conducted using a high-performance RTX-4090 GPU, taking approximately 3 hours to gather a complete dataset for each task. The exact process is detailed in Figure 7.

## F    LIMITATIONS AND DISCUSSIONS

A limitation of BARS is its applicability to non-region-based safe offline RL approaches. Our method removes the behavior regularization for states in the unsafe region. However, for settings that allow cancellation of the cost (e.g., where both positive and negative costs are permitted), it becomes challenging to define well-separated safe and unsafe regions. Thus, addressing the behavior regularization issue in such cases requires further consideration.

# G    LEARNING CURVES

The learning curves for the experiments reported in Table 1 are shown in Figure 8.

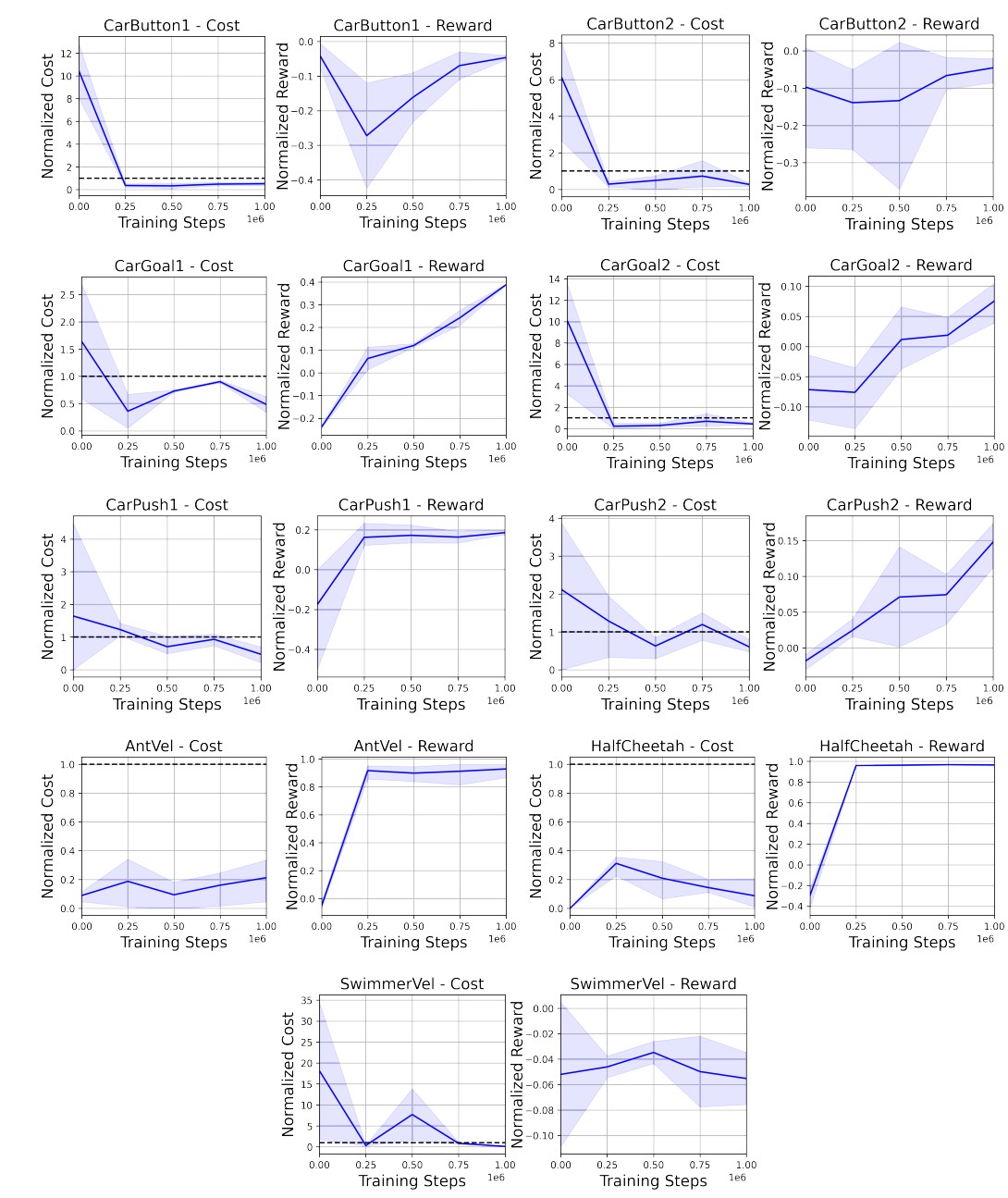

Figure 8: Learning Curves