# OpenReview forum: "Rethinking Behavior Regularization in Offline Safe RL: A Region-Based Approach"
_ICLR.cc/2025/Conference — Submitted to ICLR 2025_

### Official Review · Reviewer_WdBS · 2024-10-24

**Soundness:** 1
**Presentation:** 2
**Contribution:** 1
**Rating:** 3
**Confidence:** 5

**Summary:**

This paper proposes a safe offline RL method called BARS. The proposed method basically adds incremental modification to FISOR [1] and contains a number of theoretical flaws. The general framework and practical implementation are almost the same as FISOR. The only change is actually adding an additional term in the reward-maximizing objective, which can be ill-behaved. For detailed comments, please refer to the strengths/weaknesses.


[1] Zheng, Y. et al. Safe offline reinforcement learning with feasibility-guided diffusion model. ICLR 2024.

**Strengths:**

- The paper is easy to read.
- The reported results look good.

**Weaknesses:**

The paper lacks novelty and contains many technical issues:
- The paper shares too many similarities with FISOR, including paper structure, overall framework, and diffusion-based policy extraction. More specifically,
  - The general framework (3) of decoupling safe offline RL to maximize reward in feasible regions and minimize cost in infeasible regions is the same as FISOR.
  - One change claimed by the author is to remove the behavior regularize $D(\pi || \pi_{\beta})$ in the cost value minimization in the infeasible region. However, the authors use the same policy extraction objective for infeasible regions as in FISOR, i.e., Eq. (12), this actually corresponds to $\pi^*\propto \exp(-\mu_2 A_c(s,a))\pi_{\beta}(a|s)$, which is exactly the closed-form optimal solution of $\max_\pi -V_c^\pi$, s.t., $D_{KL}(\pi || \pi_{\beta})$. Hence the authors actually **used** implicit behavior regularization in their infeasible region optimization objective, which contradicts with their claim and motivation.
  - The authors argued that the FISOR used HJ reachability to identify the feasible region, which has several issues (L233-243). However, in their algorithm, they used the **same** reverse expectile regression objective for feasible region identification method as in FISOR (Eq.(6-7)). The only difference is that they changed the symbol of discount factor $\gamma$, cost value $h$ in the FISOR paper to $\alpha$ and $c$ in this paper. How can you claim that you have addressed the limitation in FISOR, but still used exactly the same scheme?
  - The introduction of a combined loss with two additional weight terms $b_1, b_2$ is entirely unnecessary, since it is equivalent to scaling the temperature parameter $\mu_1, \mu_2$. If you remove these two redundant hyperparameters, the policy extraction objective reduce to exactly the same as the one used in FISOR.
  - The only different design in the paper that I find different from FISOR is the additional term $Q_r^{\pi}/Q_c^{\pi}\cdot \mathbb{1}(Q_c^{\pi}>\delta)$ added to the reward maximization objective (Eq. (8)) in the feasible region. However, this is also highly problematic and lacks theoretical support. As $Q_c^{\pi}$ appears in the denominator, if $Q_c^{\pi}$ is small or takes 0, this term will explode. Since you are operating in the feasible region, ideally, the $Q_c^{\pi}$ should be 0 or a very small value, hence the optimization problem in the feasible region can be highly unstable and ill-conditioned. If one wants to avoid this, dense, non-zero cost signals need to be used, which severely limits the applicability of this method to many practical safe offline RL problems. As in many cases, we might only have sparse 0 or 1 cost signals to indicate safe/unsafe conditions, especially in complex problems where cost functions are hard to specify.
   - In summary, most parts of the algorithm actually replicate the original FISOR algorithm, while the only change is problematic.
- For the toy example in Figure 1, note that FISOR uses weighted behavior cloning ($\pi^*=\pi_{\beta}(a|s) \cdot w(s, a)$, with the weight $w$ take different forms for feasible and infeasible regions) to extract the policy. This means that if we have a higher proportion of unsafe data, we practically need to adjust the temperature parameters ($\alpha_1, \alpha_2$ in the FISOR paper, $\mu_1, \mu_2$ in this paper) to reduce the weight on the unsafe region, otherwise the imitative nature of policy extraction will cause too much learning on the unsafe data. Actually, this is exactly what the author did in their paper, they essentially introduced different loss weights $b_1, b_2$ (equivalent to adjusting $\mu_1, \mu_2$) to different tasks. In other words, the case motivated by the authors in Figure 1 can be entirely addressed by tuning the hyperparameter in FISOR.
- The use of the "normalized divergence metric" $\bar{D}$ in Eq. (5) is problematic. If you look at most offline RL literature, the behavior regularization is imposed for each state, i.e., $D(\pi(a|s) || \pi_{\beta}(a|s))$ for each $s$, and the reward value maximization is also conducted for the given state. Here, the state is the input to the policy, and the $\pi(a|s)$ is a probability distribution over the action. You can only compute KL by integrating the action but not on states. No one used the $\sum_s D(\pi(a|s) || \pi_{\beta}(a|s))$ as the behavior regularization. If you do this, then the behavioral regularization will be the same for all states when you optimize $\max_{\pi} V_r^{\pi}(s)$ for a state $s$. The proof of Theorem 3.3 (Appendix C) is thus wrong due to this error. The $D(\pi || \pi_{\beta})\leq \epsilon$ in Eq. (16) is defined for each state, but the author mistakenly treats it as the sum of KL-divergence over all states. This makes Theorem 3.3 flawed and hardly provides any theoretical support for the proposed method.

**Questions:**

- What do you mean by "learning the **least safe** region"? Any non-safe region can be considered as the "least safe" region. To guarantee safety while maximizing the reward, we actually need to identify the largest feasible region.
- The authors use different loss weights $b_1, b_2$ for different tasks, basically equivalent to adjusting the temperature hyperparameter $\mu_1, \mu_2$ for different tasks. What are the results if you use only one set of hyperparameters? Note that FISOR only uses one set of hyperparameters in all experiments. If you tune the hyperparameter for each task, for sure you will get better results.

---

> ### Author Response · Authors · 2024-11-21
> **Response to Reviewer WdBS---Part 1**
>
> We thank the reviewer for the constructive comments and feedback on our paper. We provide the detailed responses separately as follows:
>
> >**Response to Weakness 1.1:**
>
> Our work focuses on the existing issue of using behavior regularization in offline safe RL. FISOR and all the other baselines cannot ensure a safe policy when the behavior policy is suboptimal. Please see our detailed response in **[General Response](https://openreview.net/forum?id=GVhfWu5L8D&noteId=wLbYJJ4AuL)**.
>
> ---
>
> >**Response to Weakness 1.2:**
>
> We believe the reviewer may have misunderstood the TRPO used in Eq.(12). In our algorithm, we introduce the parameter $K_{\text{trust}} $, which indicates the point after which we optimize the policy using $\text{Eq.(11)}$ and $\text{Eq.(12).}$ This ensures that the current policy does not diverge significantly from the previous one by considering the trust region. This occurs **only** when the policy is safe enough. **Besides, we would like to emphasize that only using Eq.(11) can still ensure learning a safe policy (as shown in Table 4).**
>
> In our simulation, we simply set $ K_{\text{trust}}$ to half of the total training episodes without any tuning. Additionally, we would like to clarify that in $\text{Eq.(12)}$, we do not use behavior regularization from the offline dataset; instead, we employ the previously trained policy (target policy) for regularization. The parameter $ k $specifies how often we update the target policy, analogous to the target network updates used in DQN. We have clarified these points in our revision. This approach does not violate our formulation.
>
> ---
>
> >**Response to Bellman equation:**
>
> In the paper, we identified a typo in the Bellman equation (we missed the discount factor). The correct form is:
> $$
> \mathcal{L}_{\check{Q}_c} = \mathbb{E}\_{(s,a,s') \sim \mathcal{D}} \left[ \left( (1-\alpha)c(s,a) + \alpha(c(s,a) + \gamma \check{V}_c(s')) - \check{Q}_c(s,a) \right)^2 \right]
> $$
> The term $ (1-\alpha) $ is included to model the scenario where, with probability $ 1 - \alpha $, an episode ends after the current time step (e.g., after taking a very unsafe action). It is easy to show that $ c(s,a) + \gamma \check{V}_c(s') = \check{Q}_c(s,a) $ as $ \alpha \rightarrow 1 $ (we use $ \alpha = 0.99 $ in our simulation). Importantly, FISOR defines the feasible region using the value function for the hard constraint based on Hamilton-Jacobi (HJ) reachability from control theory (hence the inclusion of the maximization term). However, this formulation applies only to **deterministic systems**, which are rarely encountered in most RL environments, and cannot be generalized to problems with more flexible cost limits. As we indicated in our paper, the Bellman equation used in FISOR to train the value function does not hold in stochastic systems.
>
> ---
>
> >**Response to Weakness 1.4:**
>
> Here, $ b_1 $ and $ b_2 $ act as loss ratio controls, while $ \mu_1 $ and $\mu_2 $ serve as temperature parameters to regulate behavior regularization. Introducing $b_1 $ and $ b_2 $ allows us to easily adjust the focus on the safe or unsafe loss components.
>
> ---
>
> >**Response to Weakness 1.5:**
>
> Here, the value of the threshold $ \delta $ is the same as the cost limit $ l $ in our simulation, which means $ \delta = l > 1, l = 10, 20, 40 $. Therefore, the denominator issue does not arise (when $ Q^{\pi}_c(s,a) > \delta $, it cannot be small or close to zero), and the existing setup is adequate.
>
> ---
>
> >**Response to Weakness 1.6:**
>
> Our work focuses on the existing issue of using behavior regularization in offline safe RL. FISOR and all the other baselines cannot ensure a safe policy when the behavior policy is suboptimal. Please see our detailed response in **[General Response](#sec-difference)**.
>
> ---
>
> >**Response to Weakness 2:**
>
> We would like to emphasize that in our experiments, we did not perform a hyperparameter search in our simulations. Instead, we used a single set of loss control parameters, $ b_1 = 1 $ and $ b_2 = 3 $, for all tasks in Table 2 and most tasks in Table 1. The only exceptions were "AntVel" and "HalfCheetah" in Table 1, where we set $ b_1 = 3 $ and $ b_2 = 1 $ to achieve higher rewards. Please see details in **[General Response](https://openreview.net/forum?id=GVhfWu5L8D&noteId=wLbYJJ4AuL)**. Actually, fine-tuning parameters is not a viable solution to the problem; the issue must be addressed at the logical level.
>
> ---

---

> ### Author Response · Authors · 2024-11-21
> **Response to Reviewer WdBS---Part 2**
>
> >**Response to Weakness 3:**
>
> We believe the reviewer may have misunderstood the concept of behavior regularization. The purpose of behavior regularization is to minimize the distance between the learned policy and the behavior policy. In general, this distance should be minimized for **all** states, not just a single state. If an offline dataset is available, the loss can be expressed as:
> $$
> \mathcal{L} = \mathbb{E}\_{(s, a) \sim \mathcal{D}} \left[ D(\pi(a|s) \Vert \pi_{\beta}(a|s)) \right],
> $$
>
> where $ \mathcal{D} $ is the offline dataset of state-action pairs, and $ D(\cdot \Vert \cdot) $ represents a divergence measure (e.g., KL divergence).
>
> Since we consider a general optimization problem, the expectation should be taken over a uniform distribution of all states. Minimizing the distance for only a single state, as suggested, is incorrect because when we say two policies are close, we mean they are close across all states.
>
> The theorem can be easily generalized to the case when the initial state distribution is not fixed. Thus, our theoretical result is correct.
>
> ---
>
> >**Response to Q1:**
>
> The term "least region" here is intended to mean "largest." Specifically, we refer to the largest safe region, and all BARS does in this part is to learn and identify that largest safe region. We have clarified this in the revised paper.
>
> ---
>
> >**Response to Q2:**
>
> Same as **Response to Weakness 2.** Please see **[General Response](https://openreview.net/forum?id=GVhfWu5L8D&noteId=wLbYJJ4AuL)**. Actually, fine-tuning parameters is never a preferred solution to the problem. To further address the reviewer's concern, we did some additional simulations with tuning the parameters of FISOR. In the original FISOR, weights are split into "safe weight" and "unsafe weight." Following the reviewer's suggestion, we experimented by setting a higher ratio for unsafe weight (1:3 for "safe weight" to "unsafe weight") and increasing the cost temperature parameter to see if these adjustments would enhance FISOR’s performance. However, our experiments on CarButton1, CarButton2, and CarGoal2 (shown in Tables [CarButton1 Results](#tab-cost-reward-button1), [CarButton2 Results](#tab-cost-reward-button2), and [CarGoal2 Results](#tab-cost-reward-goal2)) indicated that FISOR was still unable to learn a safe policy when data quality was poor, even when we adjusted the parameters.
>
> Furthermore, the FISOR authors themselves state in their appendix (Section E), “However, we find that FISOR is not very sensitive to hyperparameter changes,” suggesting that parameter tuning would not significantly affect the results. Therefore, we conclude that while FISOR performs well on the standard DSRL benchmark [1], it struggles to learn a safe policy in the presence of low-quality data—a limitation observed across all baseline algorithms, not only FISOR. BARS is the first approach to effectively address this issue without requiring extensive hyperparameter tuning.
>
> #### Table: CarButton1 Results
> | **Cost Temperature** | **Normalized Cost** | **Normalized Reward** |
> |-----------------------|---------------------|-----------------------|
> | T=9 (Data Ratio 1:1)  | 2.255              | 0.008                |
> | T=5 (Data Ratio 1:1)  | 1.375              | 0.03                 |
> | T=9 (Data Ratio 1:3)  | 3.305              | 0.06                 |
> | T=5 (Data Ratio 1:3)  | 2.265              | 0.04                 |
>
> #### Table: CarButton2 Results
> | **Cost Temperature** | **Normalized Cost** | **Normalized Reward** |
> |-----------------------|---------------------|-----------------------|
> | T=9 (Data Ratio 1:1)  | 2.51               | 0.01                 |
> | T=5 (Data Ratio 1:1)  | 1.75               | -0.02                |
> | T=9 (Data Ratio 1:3)  | 6.92               | 0.02                 |
> | T=5 (Data Ratio 1:3)  | 4.425              | -0.17                |
>
> #### Table: CarGoal2 Results
> | **Cost Temperature** | **Normalized Cost** | **Normalized Reward** |
> |-----------------------|---------------------|-----------------------|
> | T=9 (Data Ratio 1:1)  | 2.11               | 0.09                 |
> | T=5 (Data Ratio 1:1)  | 2.61               | 0.12                 |
> | T=9 (Data Ratio 1:3)  | 2.28               | 0.13                 |
> | T=5 (Data Ratio 1:3)  | 1.43               | 0.14                 |

---

> > ### Comment · Reviewer_WdBS · 2024-11-24
> > **Thanks for the response, but most of my previous concerns still remain**
> >
> > I thank the authors for the detailed response, however, most of my previous concerns still remain. Therefore, I choose to keep my score unchanged.
> >
> > Specifically, in the general response,
> > - The authors claim that "This is the first paper to address how behavior regularization constraints in safe RL can lead to infeasible solutions". This is incorrect. It was discussed in CPQ [1] back in 2022.
> >
> > [1] Constraints Penalized Q-Learning for Safe Offline Reinforcement Learning. AAAI 2022.
> >
> > - The authors also claim that FISOR can only solve the "hard constraint" setting, which is also inaccurate. It is entirely fine if you add some tolerance buffer to the cost function, and solve the problem with FISOR. Also, in my opinion, ensuring hard constraints is far more desirable in real-world applications, as the consequence of policy's constraint violation can be huge.
> >
> > - Regarding the hyperparameter response, you still used two sets of hyperparameters for your tasks. Note that FISOR just used a single set of hyperparameters for experiments. Considering the close relationship of the proposed method with FISOR, this comparison is simply not convincing.
> >
> > Regarding the response of the authors,
> >
> > - Regarding the response to W1.2, yes, you used TRPO style update after $K_{trust}$ (also adds extra complexity and an extra hyperparameter to tune), but before $K_{trust}$, you are still using an AWR-style policy learning objective (Eq.(10)). This corresponds to an implicit behavior regularization as I mentioned in my comments, which contradicts with the claim and motivation of the authors that behavior regularization should be removed for optimization in the unsafe region. Moreover, if the idea of removing the behavior regularization in the unsafe region really works, why do you still need to add a second TRPO-type update stage in your policy learning?
> >
> > - Regarding the response to W1.4, I'm not convinced by this claim. If set appropriately, solely adjusting $\mu_1$ and $\mu_2$ can achieve the same effect.
> >
> > - Regarding the response to W1.5, OK, but adding a value ratio will introduce noticeable non-smoothness around the boundaries of $Q_c^{\pi}(s, a)>\delta$ in your learning objective, potentially causing learning stability issues.
> >
> > - Regarding the response to W2, please check my comments on the general response. I would encourage the author to report all of their results using a single set of hyperparameters, as in FISOR.
> >
> > - Regarding the response to W3, I'm not convinced by the argument of the authors. Please name a single well-known offline RL algorithm, that uses the sum over all states on KL-divergence as behavior regularization in their policy optimization objective, as what is used in the theoretical analysis of this paper.
> >
> > - Regarding the response to Q1, why not simply call it the "largest safe region", rather than still using the confusing "least safe region" in your revised paper?

---

> > > ### Author Response · Authors · 2024-11-25
> > > **Response to WdBS---New response Part1**
> > >
> > > >**Q1:**
> > >
> > > First, we believe the reviewer may have misunderstood the main problem we aim to address. Our focus is on scenarios where poor data quality causes behavior regularization applied uniformly across all states to fail in learning a safe policy. While CPQ [R1] mentions behavior regularization, they neither provide a mathematical formulation nor a solution to this issue, apart from adding a penalty term during Q-function updates. Moreover, they do not present any experimental evidence to discuss this problem.
> > >
> > > [R1] Constraints Penalized Q-Learning for Safe Offline Reinforcement Learning. AAAI 2022.
> > >
> > > ---
> > >
> > > >**Q2:**
> > >
> > > As stated in our response and noted by other reviewers, FISOR is a cost-limit-agnostic algorithm. While we respect your opinion, we must point out that hard constraints are a specific case of the average constraint setting. Additionally, as we mentioned, BARS is capable of handling both cases.
> > >
> > > ---
> > >
> > > >**Q3&Q7:**
> > >
> > > First, note that in the main experiment in Table 2, we used only one set of hyperparameters ($b_1=1, b_2=3$) to train our algorithm and report the results. The results in Table 1 are intended to demonstrate that when the data quality is good, BARS can also learn a good policy that achieves an appropriate reward compared to existing methods. **We never claimed that our goal was to design a new algorithm that beats all existing methods under standard benchmarks—did we say this?** Furthermore, if we exclude the experiments using ($b_1=3, b_2=1$) in "AntVelocity" and "HalfCheetah" in Table 1, the experimental results are as follows:
> > >
> > > | **Task**      | **FISOR Reward** | **FISOR Cost** | **BARS Reward** | **BARS Cost** |
> > > |---------------|------------------|----------------|-----------------|---------------|
> > > | CarButton1    | -0.11            | 0.73           | -0.04           | 0.51          |
> > > | CarButton2    | -0.001           | 0.83           | -0.04           | 0.27          |
> > > | CarGoal1      | 0.42             | 1.63           | 0.38            | 0.48          |
> > > | CarGoal2      | 0.03             | 0.24           | 0.07            | 0.44          |
> > > | CarPush1      | 0.24             | 1.61           | 0.18            | 0.47          |
> > > | CarPush2      | 0.10             | 1.21           | 0.148           | 0.64          |
> > > | SwimmerVel    | -0.04            | 0.31           | -0.05           | 0.125         |
> > > | **Average**   | 0.091            | 0.93           | 0.092           | 0.42          |
> > >
> > > *Table 1: Performance comparison between FISOR and BARS across different tasks.*
> > >
> > > ---
> > >
> > > >**Q4:**
> > >
> > > First, we clearly state that setting $K_{\text{trust}}$ to half the total training episodes is done **without tuning**. Additionally, we have consistently pointed out that **we never use behavior regularization between the current policy and the behavior policy $\pi_{\beta}$ from the offline dataset**. We are unsure why the reviewer continues to claim that this "contradicts the authors' assertion that behavior regularization should be removed for optimization in the unsafe region." We have clearly stated the removal of behavior regularization for the behavior policy $\pi_{\beta}$. The reason we added TRPO-style updating is to improve BARS's performance in **achieving higher rewards**, which is unrelated to safety considerations. **Using Eq. (10) alone is sufficient to learn a safe policy across all experiments in Table 1 and Table 2.** To further validate our approach, we conducted an experiment (Table 2) in the CarButton2 environment (data ratio 1:1 in Table 2) to demonstrate that our key components are essential for learning a safe policy under different data distributions. In this experiment, we divided the following four situations and reported the results in Table 2:
> > >
> > > - (i) Use ${D}(\pi||\pi_{\beta})\le\epsilon$ only in safe region (**Eq.(10)**).
> > > - (ii) Use ${D}(\pi||\pi_{\beta})\le\epsilon$ both in safe and unsafe region.
> > > - (a) Use $\frac{Q\_r^\pi(s,a)}{Q\_c^\pi(s,a)} \cdot 1\_{\{Q\_c^\pi(s,a) > \delta \}} + Q\_r^\pi(s,a) \cdot 1\_{\{Q\_c^\pi(s,a) \leq \delta \}}$ (**Eq.(7)**).
> > > - (b) Use ${Q\_r^\pi(s,a)} \cdot 1\_{\{Q\_c^\pi(s,a) > \delta \}} + Q\_r^\pi(s,a) \cdot 1\_{\{Q\_c^\pi(s,a) \leq \delta \}}$.
> > >
> > > The experimental results showed that using (ii) cannot assure a safe policy; component (i) is the crucial factor to guarantee a safe policy. This experiment highlights how each component of BARS contributes to learning a safer policy.
> > >
> > > | **Component**          | **Normalized Cost** | **Normalized Reward** |
> > > |------------------------|---------------------|-----------------------|
> > > | Using (ii), (b)        | 2.69                | -0.05                 |
> > > | Using (i), (b)         | 0.785               | -0.07                 |
> > > | Using (ii), (a)        | 1.43                | -0.01                 |
> > > | Using (i), (a) (BARS)  | 0.63                | -0.08                 |
> > >
> > > *Table 2: Component Analysis*
> > >
> > > ---

---

> > > > ### Author Response · Authors · 2024-11-25
> > > > **Response to WdBS---New response Part2**
> > > >
> > > > >**Q5:**
> > > >
> > > > We acknowledge that the parameters inside the exponential term and those controlling the ratio between safe and unsafe loss have different sensitivities. However, we do not agree that this constitutes a weakness of BARS, as it is unrelated to the contributions claimed in our paper.
> > > >
> > > > ---
> > > >
> > > > >**Q6:**
> > > >
> > > > As demonstrated in our simulation results, there are clearly no stability issues. We strongly disagree with the reviewer's claim, which appears to be based on speculation rather than evidence.
> > > >
> > > > ---
> > > >
> > > >
> > > > >**Q8:**
> > > >
> > > > We suggest the reviewer to understand more about the KL divergence between two distributions and how to control the distribution shift in RL. A good survey is [R1].
> > > >
> > > > [R1] Levine, Sergey, et al. "Offline reinforcement learning: Tutorial, review, and perspectives on open problems."
> > > >
> > > > ---
> > > >
> > > > >**Q9:**
> > > >
> > > > We are unsure why the use of this word continues to bother the reviewer. Our use of the term "least" originates from the phrase "least (largest) safe region," which refers to a region where the agent can find at least one policy that reaches the destination without violations. We do not see any reason why this term cannot be used, especially since we have clearly explained the meaning of this region in the section "Learning the least safe region."

---

> > > ### Comment · Reviewer_LCMx · 2024-11-29
> > >
> > > Hi, Reviewer WdBS
> > >
> > > I would like to know if the author's response has resolved your remaining concerns. If there are any remaining questions, I hope you can raise them as soon as possible.

---

> > > > ### Comment · Reviewer_WdBS · 2024-12-02
> > > > **Response to Reviewer LCMx**
> > > >
> > > > The authors have not satisfactorily addressed many of my previous concerns. As also pointed out by other reviewers, the paper has quite many issues, ranging from overclaiming, similarity to existing works, mismatched design motivation and the final practical algorithm, problematic theoretical analysis (the authors' responses also reflect their lack of in-depth understanding of offline RL research), problems with the reachability modeling and reported results (mentioned by other reviewers). Therefore, I choose to keep my evaluation unchanged.

---

### Official Review · Reviewer_RHPd · 2024-10-25

**Soundness:** 2
**Presentation:** 2
**Contribution:** 2
**Rating:** 3
**Confidence:** 4

**Summary:**

BARS is developed to tackle challenges in offline safe reinforcement learning by applying selective behavior regularization according to the safety of states. This method shows improved performance in terms of safety and rewards compared to existing approaches, especially in scenarios involving suboptimal behavior policies and low-quality datasets.

**Strengths:**

- Learning a safe policy from an offline dataset is important and useful in real-world applications.
- Ensuring that every state meets safety requirements has practical needs.

**Weaknesses:**

## Key components lack theoretical analysis and proofs

- The author removes distribution constraints in unsafe areas to avoid limits that hinder obtaining safe actions. However, this might lead to OOD issues, which could still result in unsafe actions. How to resolve this remains an issue.
- It's unclear whether Equation 7 is the Hamilton-Jacobi (HJ) value function or the cost value function. If it is the cost value function, not having a discount factor could undermine the Bellman iteration, which needs discounting for convergence. If it's the HJ value function, it does not align with its usual definition[1]. Also, if it is the HJ function, how it ensures safety with different cost limits is not clear, given that HJ usually focuses on zero constraint violations. These issues suggest there might be errors in the training algorithm's formulation.
- It's unclear how Equation 5 transforms into Equation 9. Additionally, the training loss in Equation 12 lacks theoretical backing. Although it resembles the idea used in TRPO, it's uncertain if this approach is correct in a safe offline setting.

[1] Fisac, Jaime F., et al. "Bridging hamilton-jacobi safety analysis and reinforcement learning." *2019 International Conference on Robotics and Automation (ICRA)*. IEEE, 2019.

## Practical application issues

The training of the cost value function and the value function requires diffusion policy sampling, which is time-consuming. Additionally, the train-collect-train framework is problematic as it offers no safety guarantees during the collection procedure. This is in conflict with the goal of offline safe learning.

**Questions:**

- The largest feasible region shown in Figure 1 should not be related to data quality, yet it appears different in panels b, c, and d. Why?
- The cost threshold in the code is set to 8. How was this value determined?
- It's strange that in Table 3, fisor achieves different performances under different cost limits. fisor should consider hard constraints, meaning its performance should be independent of the cost limit, right?

---

> ### Author Response · Authors · 2024-11-21
> **Response to Reviewer RHPd---Part 1**
>
> We thank the reviewer for the constructive comments and feedback on our paper. We provide the detailed responses separately as follows:
>
> >**Response to OOD issue:**
>
> OOD actions can be managed by applying behavior regularization in the safe region. In the unsafe region, we directly minimize the cost function, which is trained using IQL. We thank the reviewer for raising this question, as it highlights the potential for further improvement by incorporating a penalty term into the critic network (similar to CPQ) to enhance control over OOD actions.
>
> ---
>
> >**Response to Bellman equation:**
>
> In the paper, we identified a typo in the Bellman equation (we missed the discount factor). The correct form is:
> $$
> \mathcal{L}_{\check{Q}_c} = \mathbb{E}\_{(s,a,s') \sim \mathcal{D}} \left[ \left( (1-\alpha)c(s,a) + \alpha(c(s,a) + \gamma \check{V}_c(s')) - \check{Q}_c(s,a) \right)^2 \right]
> $$
> The term $(1-\alpha) $ is included to model the scenario where, with probability $ 1 - \alpha $, an episode ends after the current time step (e.g., after taking a very unsafe action). It is easy to show that $ c(s,a) + \gamma \check{V}_c(s') = \check{Q}_c(s,a) $ as $ \alpha \rightarrow 1 $ (we use $ \alpha = 0.99 $ in our simulation). Importantly, FISOR defines the feasible region using the value function for the hard constraint based on Hamilton-Jacobi (HJ) reachability from control theory (hence the inclusion of the maximization term). However, this formulation applies only to **deterministic systems**, which are rarely encountered in most RL environments, and cannot be generalized to problems with more flexible cost limits. As we indicated in our paper, the Bellman equation used in FISOR to train the value function does not hold in stochastic systems.
>
> ---
>
> >**Response to Weakness 3:**
>
> The transformation from Eq. (5) to Eq. (9) involves:
> 1. Eq. (9) introduces $ A_r^\pi(s, a) $ and $\check{A}_c(s, a) $ to explicitly measure the impact of actions on the optimization objective.
> 2. The constraint $ \int_a \pi(a|s) \, da = 1 $ ensures that the policy $ \pi(a|s) $ is a valid probability distribution over actions for each state $ s $.
> 3. Instead of using $ V_c^\pi(s) \leq l $ as in Eq. (5), Eq. (9) refines this by constraining individual actions $ \\{a | \check{Q}_c(s, a) \leq l \\}$, ensuring that every action chosen adheres to the cost limit.
>
> ---

---

> ### Author Response · Authors · 2024-11-21
> **Response to Reviewer RHPd---Part 2**
>
> >**Response to Practical application issues:**
>
> **Time-consuming:** We would first like to acknowledge FISOR for providing an approach that reduces training costs. Their novel energy-guided sampling method simplifies the training process by eliminating the need for a complex time-dependent classifier. Additionally, our implementation leverages the JAX framework, which offers a significant speed advantage compared to other frameworks like PyTorch and TensorFlow. In practice, training for 1 million steps takes approximately 50 minutes on a single RTX 4090 GPU, enabling high efficiency and scalability (Table [Training Steps/Minutes](#tab-training-time)).
> #### Table: Training Steps/Minutes
> |                    | CPQ (PyTorch) | COptiDICE (PyTorch) | FISOR (JAX) | BARS (JAX) |
> |--------------------|---------------|---------------------|-------------|------------|
> | Training 1e6 steps/mins | 200           | 80                  | 44          | 53         |
>
> **No safety guarantees:** Regarding the "train-collect-train" framework, we highlight the improved quality of the new data, demonstrating that our policy adheres to safety requirements, as the mean episode cost is consistently lower than the cost limit 10.
>
> #### Table: Episode and Total Costs Comparison
> | Dataset            | **Mean Episode Cost** | **Total Cost** |
> |--------------------|-----------------------|----------------|
> | Offline Dataset    | 69.72                | 33182          |
> | First collect data | 7.45                 | 4216           |
> | Second collect data| 4.41                 | 3724           |
>
> ---
>
> >**Response to Q1:**
>
> Since the regions are plotted based on the trained $ \check{V}_c$, and (b), (c), and (d) use three different networks, it is expected that the plotted regions may show minor differences.
>
> ---
>
> >**Response to Q2:**
>
> The value "8" is simply a static configuration in the file. When we uploaded the file, we did not provide a complete running script. In practical training, we set $ \delta = l = \text{cost limit} $.
>
> ---
>
> >**Response to Q3:**
>
> FISOR is designed to be cost limit-agnostic due to its reliance on a hard constraint framework. However, as we indicated in our paper, this design limits the generalization of their approach to other scenarios where a specific cost limit is considered, which is a more practical setting. Our aim is to demonstrate that increasing the cost limit should potentially lead to better performance for an algorithm. We do not understand why the reviewer considers this to be a weakness of our paper, especially since FISOR is one of the most closely related works and it is reasonable to add more discussion.

---

> > ### Comment · Reviewer_RHPd · 2024-11-23
> >
> > Thank you for the author's response. However, some questions remain:
> >
> > - The author revised Equation (6) in the paper. Based on the latest version, I believe this now corresponds to the standard VC Bellman iteration formula, rather than the value function defined under HJ [1]. Regarding VC, it has already been widely applied in the field of safe offline RL, yet the author does not provide sufficient discussion on this aspect. Moreover, alpha is a terminal flag that is typically set to 0 or 1. What is the rationale behind the author’s choice of 0.99? Lastly, in line 269, the author claims that alpha → 1 satisfies the contraction mapping, but the correct condition should be gamma → 1 to ensure the contraction mapping of the Bellman operator.
> > - The author also needs to provide more discussion and analysis regarding the practical feasibility of the train-collect-train framework. While this framework might be reasonable for offline RL or offline-to-online RL, it raises concerns when safety constraints are considered. Ignoring safety during the data collection phase is problematic, as it introduces significant risks, even if the experiments suggest that the algorithm does not violate the cost limit. From a real-world application perspective, this approach presents substantial safety hazards.
> > - The author claims that the model can adapt to various cost limit scenarios. However, BARS still requires retraining under different cost limit settings. Compared to existing models [2] that can handle multiple cost limits with a single model, BARS does not demonstrate a clear advantage. Additionally, I agree with Reviewer Yqdv that it is necessary to compare BARS with CDT [2], especially since the author claims that their method is designed for scenarios involving dynamic cost limits.
> >
> > [1] Fisac, Jaime F., et al. "Bridging Hamilton-Jacobi safety analysis and reinforcement learning." 2019 International Conference on Robotics and Automation (ICRA). IEEE, 2019.
> > [2] Zuxin Liu, et al. "Datasets and benchmarks for offline safe reinforcement learning." arXiv preprint arXiv:2306.09303 (2023).

---

> > > ### Author Response · Authors · 2024-11-24
> > > **Response to RHPd---New response**
> > >
> > > >**Q1:**
> > >
> > > First, we want to correct a misunderstanding: the statement "alpha is a terminal flag that is typically set to 0 or 1" is incorrect. In [R3], the discount factor $ \alpha \in [0,1) $, meaning that $ \alpha $ cannot be set to 1. The choice of $ \alpha = 0.99 $ follows the original FISOR parameter settings. As mentioned previously, we did not conduct an extensive parameter fine-tuning process. The use of $ 1-\alpha $ represents the probability that an episode may terminate after this step with the probability $ 1-\alpha $. We adapt the Discounted Reach-Avoid Bellman Equation [R1, R2] to the standard Bellman Equation setting for stochastic environments. We believe that the formulation with an absorbing probability is more practical. While we do not observe significant differences between using Eq. (6) and the standard Bellman equation in our simulations (as we set $ \alpha = 0.99 $), we think this approach has potential applications in environments where episodes can terminate early due to absorbing states.
> > >
> > > [R1] Hsu, Kai-Chieh, Vicenç Rubies-Royo, Claire J. Tomlin, and Jaime F. Fisac. "Safety and liveness guarantees through reach-avoid reinforcement learning." arXiv preprint arXiv:2112.12288 (2021).
> > >
> > > [R2] So, Oswin, Cheng Ge, and Chuchu Fan. "Solving Minimum-Cost Reach Avoid using Reinforcement Learning." arXiv preprint arXiv:2410.22600 (2024).
> > >
> > > [R3] Fisac, Jaime F., et al. "Bridging Hamilton-Jacobi safety analysis and reinforcement learning." 2019 International Conference on Robotics and Automation (ICRA). IEEE, 2019
> > >
> > > ---
> > >
> > > >**Q2:**
> > >
> > > First, we would like to clarify the purpose of this framework: when data quality is poor (as in the setting in Table 2), we first train a safe policy, then use this policy to collect new data and continue training on higher-quality data. Our goal is to demonstrate that this process can gradually improve the agent's performance. The analysis based on our newly collected data clearly shows that the second collection has significantly lower average episode cost and total cost compared to the offline dataset and the first collection. While achieving zero violations is an interesting question for future exploration, it is not the primary focus of this paper.
> > >
> > > |                         | **Mean Episode Cost** | **Total Cost** |
> > > |-------------------------|-----------------------|----------------|
> > > | Offline Dataset         | 69.72                 | 33182          |
> > > | First collected data    | 7.45                  | 4216           |
> > > | Second collected data   | 4.41                  | 3724           |
> > >
> > > *Table 1: Episode and Total Costs Comparison*
> > >
> > > ---
> > >
> > > >**Q3:**
> > >
> > >  First, we want to clarify the purpose of this different cost limit experiment: BARS is more adaptable to practical scenarios where varying cost limits are required. Here, we provide CDT's performance on the task "CarButton1" (cost limit $ l = 10 $) and will include more results of CDT in our final revision. However, it is worth noting that, as demonstrated in FISOR (Table 1), CDT (with a cost limit of 10, which is the same as our setting in Table 1) fails to learn a safe policy for most tasks. In [R1], CDT also fails to learn a safe policy under cost limits of 20, 40, and 80 for the tasks 'CarButton1,' 'CarButton2,' 'CarGoal1,' and 'CarGoal2.' Again, we would like to emphasize that our main contribution focuses on cases where the behavior policy is suboptimal. As we showed in Table 2 of our paper, the results in Table 1 aim to demonstrate that BARS can achieve better performance even when the behavior policy is good enough.
> > >
> > > [R1] Liu, Zuxin, Zijian Guo, Haohong Lin, Yihang Yao, Jiacheng Zhu, Zhepeng Cen, Hanjiang Hu et al. "Datasets and benchmarks for offline safe reinforcement learning." arXiv preprint arXiv:2306.09303 (2023).

---

> > > > ### Comment · Reviewer_RHPd · 2024-11-25
> > > >
> > > > Thank you for your response.
> > > >
> > > > The author notes, “In [R3], the discount factor alpha is in [0,1), meaning that alpha cannot be set to 1,' and 'The use of 1-alpha represents the probability that an episode may terminate after this step with the probability 1-alpha .' Could you please clarify whether alpha is intended as a discount factor or a termination flag? It appears there might be some confusion on this point. Additionally, the author mentions, “Here, we provide CDT's performance on the task "CarButton1" (cost limit $ l = 10 ").” Could you provide the results?
> > > >
> > > > Overall, this paper has many theoretical issues and contradictions, and many of its designs lack theoretical justification. I will maintain my score.

---

> > > > > ### Author Response · Authors · 2024-11-25
> > > > > **Response to PHPd---Second New response**
> > > > >
> > > > > [R1] clearly state that "the discount factor $\gamma\in[0,1)$ can be seen as the probability of the episode continuing, **with $1-\gamma$ conversely representing the probability
> > > > > of transitioning to a terminal state**". Besides, we provide CDT performance on the task "CarButton1"(cost limit=$ l = 10 $) and will include the results of CDT in our final revision. However, it is worth noting that, as demonstrated in  FISOR (Table 1), CDT (with a cost limit of 10, which is the same as ours' set in Table 1) fails to learn a safe policy for most tasks.
> > > > >
> > > > > | **Task**      | **Reward** | **Cost**   |
> > > > > |---------------|------------|------------|
> > > > > | CarButton1    | 0.26       | **8.326**  |
> > > > >
> > > > > *Table : CDT Performance*
> > > > >
> > > > > [R1] Fisac, Jaime F., et al. "Bridging Hamilton-Jacobi safety analysis and reinforcement learning." 2019 International Conference on Robotics and Automation (ICRA). IEEE, 2019

---

> > > > > > ### Comment · Reviewer_RHPd · 2024-12-01
> > > > > >
> > > > > > Thank you for the author’s response. However, I still think there may be a theoretical issue:
> > > > > >
> > > > > > - The author has referenced literature related to hj reachability to discuss the correctness of equation (6). However, by expanding the parentheses and combining like terms, it becomes evident that the middle part of equation (6) is actually c + alpha gamma Vc, which is essentially a cost value function under a conventional definition, with the only difference being that the discount factor is 0.99 * 0.99. While this definition is theoretically sound, the choice to define it as alpha * gamma feels somewhat unusual.
> > > > > > - Another concern is that the cost value function used by the author assumes c>=0 [1] [2], but in practice, the author has used -1 for the safe state. This results in a biased definition of Vc <= l, and it seems challenging to derive the safe region under the cost limit l mathematically. Since Vc is central to the theory presented in the paper, I think this point needs further discussion.
> > > > > >
> > > > > > [1] Xu, Haoran, Xianyuan Zhan, and Xiangyu Zhu. "Constraints penalized q-learning for safe offline reinforcement learning." Proceedings of the AAAI Conference on Artificial Intelligence. Vol. 36. No. 8. 2022.
> > > > > >
> > > > > > [2] Liu, Zuxin, et al. "Constrained decision transformer for offline safe reinforcement learning." International Conference on Machine Learning. PMLR, 2023.

---

### Official Review · Reviewer_Yqdv · 2024-11-02

**Soundness:** 2
**Presentation:** 2
**Contribution:** 2
**Rating:** 5
**Confidence:** 4

**Summary:**

This paper tackles challenges in offline safe RL caused by behavior regularization, which keeps the learned policy close to the behavior policy but can lead to unsafe actions when the behavior policy itself is suboptimal. The authors introduce BARS, an algorithm that distinguishes between safe and unsafe states and applies selective behavior regularization accordingly. BARS ensures safety and high performance even with low-quality datasets, outperforming current baselines in both reward and safety metrics.

**Strengths:**

(1) Interesting topics. Safety issues are important to Reinforcement Learning applications, and this paper has shed some insights on how to achieve safety in offline RL.

(2) Good insight and intuitive idea: Behavior regularization (data quality) indeed has a strong impact on the final performance of machine learning systems, as revealed in many data-centric learning works [1]

Reference:
[1] Chunting, Zhou, et al. "Lima: Less is more for alignment." NeurIPS 2023.

**Weaknesses:**

(1) Typo: Line 305, Liu et al. 2023b did not use a diffusion model to parameterize the policy network.

(2) Concerns about the experiment performance. In Table 1, some results of the proposed method raise concerns about its effectiveness. For example, in CarButton1 and SwimmerVel tasks, the rewards are negative. However, as reported in the original DSRL paper, some baselines such as CDT and Safe-BC should perform well with a high reward and low cost. The authors also did not report the comparison between these baselines.

(3) Confusion about the new setting: Safe offline RL with new Data Collection. Why did the authors discuss this part I believe this part should fall into the categories of continued learning or active learning. If the authors want to claim the contribution using this part, I believe baseline comparison for active learning/offline-to-online adaptive learning should be necessary.

(4) The proposed method shares many insights over the previous work: FISOR [1]. For example, using a diffusion model to parameterize the policy, using a weighted loss for training, and using control theory (feasibility) to help describe the algorithm and derive theorems. Thus, I suggest the authors to further clarify the difference between the proposed method and FISOR.

Reference:

[1] Yinan, Zheng,  et al. "Safe offline reinforcement learning with feasibility-guided diffusion model." ICLR 2024.

**Questions:**

(1) Could you clarify if you also used a weighted loss (Line 307) in baseline training (CPQ and COptiDICE) for a fair comparison?

(2) Can you provide more baseline results such as BCQ-Lag, CDT, and Safe-BC? As they are also implemented in the DSRL benchmark [1] you use, it would be easy to make a comparison.

(3) How do you choose the hyperparameters b1 and b2 in equation 13?

(4) Can you clarify what results you believe are not consistent with the FISOR paper as you stated in Lines 429-431?

(5) Is equation (13) the key component to mitigate the claimed regularization issues? If so, I believe more discussion about the extension of the proposed method to the general method including those that do not require diffusion policy such as BCQ-Lag and CDT is necessary. If not, I suggest the authors to further clarify and locate the core components to solve this issue.

(6) In Table 3, the authors show the experiments with different thresholds. The authors only show one experiment task and one baseline (FISOR). All the results are safe for BARS and FISOR, and the strength of the BARS is shown by higher rewards. However, based on the previous description “BARS is a novel algorithm that distinguishes between safe and unsafe states and applies region-specific, selective behavior regularization to optimize the policy.” (lines 24-26), the strength of BARS should be the ability to achieve safety under safety-critical situations. I expect the authors to further clarify the strength of BARS.


Reference:

[1] Zuxin Liu, et al. "Datasets and benchmarks for offline safe reinforcement learning." arXiv preprint arXiv:2306.09303 (2023).

---

> ### Author Response · Authors · 2024-11-21
> **Response to Reviewer Yqdv---Part1**
>
> We thank the reviewer for the constructive comments and feedback on our paper. We provide the detailed responses separately as follows:
>
> **We have carefully polished our paper and fixed the typos.**
>
> >**Response to Weakness 2:**
>
> Thanks for your suggestion. We implemented the BC-SAFE using the same dataset of CarButton1 we used in Table 2 to complete a comparison (Table [BC-SAFE Comparison](#tab-bcsafe-carbutton)). The result shows that BC-SAFE also fails to learn a safe policy. Additionally, we plan to include CDT in the final revision. However, due to the limited time, we could not complete all the simulations. Furthermore, we believe CDT is also unlikely to achieve a safe policy when the behavior policy is poor. As reported in FISOR's paper and (Liu et al., 2023a), CDT cannot guarantee a safe policy, while FISOR demonstrates the best results among current methods.
>
> #### Table: BC-SAFE Comparison
> | Task          | Data Ratio | **Normalized Reward** | **Normalized Cost** |
> |---------------|------------|-----------------------|---------------------|
> | CarButton1    | 1:1        | 0.023                | 1.47                |
> | CarButton1    | 1:3        | 0.004                | 1.24                |
>
>
> ---
>
> >**Response to Weakness 3:**
>
> The motivation for new data collection stems from the fact that the performance of offline RL algorithms is highly dependent on the quality of the dataset. A suboptimal behavior policy can limit the ability to learn the optimal policy. In offline Safe-RL, collecting new data is risky unless a reasonable and reliable safe policy is available (this is not an issue in unconstrained RL). Therefore, we propose a framework to demonstrate a potential way of gradually and **safely** learning a **safe** policy. We do not compare our approach with existing offline-to-online algorithms because our focus is not on solving the problem of pre-trained offline policies with online fine-tuning and a limited number of interactions. This represents a different setting.
>
> In the following table, we report the average cost per episode (cost limit $ l = 10$) based on $ 600,000 $ data samples with 600 trajectories for the original offline dataset, the first collection (after training for $ 200K $ steps), and the second collection (after training for $ 400K $ steps). We can clearly observe that the mean episode cost drops dramatically after each new collection, which implies that the policy is **safe enough** for exploration.
>
> #### Table: Average Episode Cost
> | Dataset               | **Mean Episode Cost** |
> |-----------------------|-----------------------|
> | Offline Dataset       | 69.72                |
> | First collect data    | 7.45                 |
> | Second collect data   | 4.41                 |
>
> ---
>
> >**Response to different as FISOR:**
>
> Our work focuses on the existing issue of using behavior regularization in offline safe RL. FISOR and all the other baselines cannot ensure a safe policy when the behavior policy is suboptimal. Please see our detailed response in **[General Response](https://openreview.net/forum?id=GVhfWu5L8D&noteId=wLbYJJ4AuL)**.
>
> ---
> >**Response to Q1:**
>
> The weighted loss function is specifically designed for diffusion settings. CPQ and COptiDICE are not diffusion-based approaches, so they cannot be directly applied with a weighted loss. Additionally, BARS incorporates several key components to ensure that the modified optimization problem can be efficiently solved. Simply adding the weighted loss to their algorithms may not work effectively.
>
> ---

---

> ### Author Response · Authors · 2024-11-21
> **Response to Reviewer Yqdv---Part2**
>
> >**Response to Q2:**
>
> Thank you for your suggestion. Given the limited time of rebuttal, we conducted two additional baselines, BCQ-Lag and BEAR-Lag, on CarGoal1 and CarGoal2 to evaluate their performance under different data distributions, as shown in Tables [BEAR-Lag Results](#tab-bearcargoal) and [BCQ-Lag Results](#tab-bcqcargoal). The results demonstrate that both baselines fail to learn a **safe policy**, further highlighting the strong performance of BARS. We will add the results for other environments in the final revision.
>
> #### Table: BEAR-Lag Results
> | Task          | Data Ratio | **Normalized Reward** | **Normalized Cost** |
> |---------------|------------|-----------------------|---------------------|
> | CarGoal1      | 8:1        | 0.66                 | 2.2                 |
> |               | 1:1        | 0.45                 | 3.27                |
> |               | 1:3        | 0.53                 | 1.6                 |
> | CarGoal2      | 8:1        | 0.32                 | 3.65                |
> |               | 1:1        | 0.25                 | 7.19                |
> |               | 1:3        | 0.26                 | 2.95                |
>
> #### Table: BCQ-Lag Results
> | Task          | Data Ratio | **Normalized Reward** | **Normalized Cost** |
> |---------------|------------|-----------------------|---------------------|
> | CarGoal1      | 8:1        | 0.51                 | 2.28                |
> |               | 1:1        | 0.44                 | 3.24                |
> |               | 1:3        | 0.38                 | 1.26                |
> | CarGoal2      | 8:1        | 0.19                 | 1.54                |
> |               | 1:1        | 0.30                 | 5.07                |
> |               | 1:3        | 0.29                 | 2.75                |
>
> ---
>
> >**Response to Q3:**
>
> In our experiments, we use the parameters provided in Appendix Table 7 and did not perform extensive tuning on these parameters. Please see the details in **[General Response](https://openreview.net/forum?id=GVhfWu5L8D&noteId=wLbYJJ4AuL)**.
>
> ---
>
> >**Response to Q4:**
>
> We refer to the result under FISOR as $ (0.268, 0.72)$ , which is slightly worse than what they reported in their paper $ (0.30, 0.35) $.
>
> ---
>
> >**Response to Q5:**
>
> Thanks for your question. We give a more clear summary of our contributions in the revised paper. The key components are:
> 1. $\text{Eq.(5)}$, where we remove the behavior regularization for the unsafe region.
> 2. The Region-Based Risk-Reward Optimization technique in $\text{Eq.(8)}$.
>
> To further validate our approach, we conducted an experiment (Table [Component Analysis](#component-analysis)) in the CarButton2 environment (data ratio 1:1 in Table 2) to demonstrate that our key components are essential for learning a safe policy under different data distributions.
>
> In this experiment, we divide the following four situations and report the results in Table [Component Analysis](#component-analysis):
>
> 1. (i) $ {D}(\pi||\pi_{\beta})\le\epsilon $ only in the safe region.
> 2. (ii) $ {D}(\pi||\pi_{\beta})\le\epsilon $ both in safe and unsafe regions.
> 3. (a) $ \frac{Q_r^\pi(s,a)}{Q_c^\pi(s,a)} \cdot 1_{\{Q_c^\pi(s,a) > \delta \}} + Q_r^\pi(s,a) \cdot 1_{\{Q_c^\pi(s,a) \leq \delta \}} $.
> 4. (b) $ {Q_r^\pi(s,a)} \cdot 1_{\{Q_c^\pi(s,a) > \delta \}} + Q_r^\pi(s,a) \cdot 1_{\{Q_c^\pi(s,a) \leq \delta \}} $.
>
> This experiment shows that using (ii) cannot assure a safe policy, and component (i) is the crucial factor to guarantee a safe policy. The results highlight how each component of BARS contributes to learning a safer policy.
>
> #### Table: Component Analysis
> | **Component**        | **Normalized Cost** | **Normalized Reward** |
> |-----------------------|---------------------|-----------------------|
> | Using (ii), (b)        | 2.69               | -0.05                |
> | Using (i), (b)        | 0.785              | -0.07                |
> | Using (ii), (a)        | 1.43               | -0.01                |
> | Using (i), (a) (**BARS**) | 0.63               | -0.08                |
>
> Our method can be applied to any region-based approach that distinguishes between safe and unsafe regions. Since BCQ-Lag and CDT are not region-based approaches, they cannot leverage this technique to enhance their performance.
>
> ---
>
> >**Response to Q6:**
>
> Thanks for the suggestion, we have made it more clear in the updated revision.

---

> > ### Comment · Reviewer_Yqdv · 2024-11-22
> > **Concerns have not been resolved**
> >
> > I thank the authors for their time and commitment. However, my concerns have not been resolved.
> >
> > 1. Regarding my previous comment (W2), I pointed out that the results in Table 1 were unsatisfactory and requested more comparisons with BC-safe. However, the authors did not directly address this comment; instead, they conducted one additional experiment in Table 2. Since these experiments are not computationally demanding, I still expect the authors to report the results of CDT and BC-safe in Table 1.
> >
> > 2. Concerning my concerns about the similarity with FISOR, the authors responded:
> >
> > > This is the first paper to address how behavior regularization constraints in safe RL can lead to infeasible solutions, particularly with multiple constraints.
> >
> > Since there are already prior works discussing behavior regularization constraints in safe RL, I expect the authors to validate their claimed contributions concerning multiple constraints by highlighting the corresponding experimental results.
> >
> > 3. In response to my previous question (Q1) about the clarification of weighted training, the authors stated that it is only designed for the diffusion model. I raised additional concerns regarding this. Could you elaborate further? Why would assigning different weights to different state-action pairs during policy training not be feasible for baseline methods such as CPQ and COptiDICE? I need further clarification.
> >
> > 4. The authors’ response to Q6 was simply: "Thanks for the suggestion; we have made it more clear in the updated revision." Could you highlight the revision? I do not see any updates to Table 3.

---

> > > ### Author Response · Authors · 2024-11-23
> > > **Response to Reviewer Yqdv---Answer New question--Q1, Q2, Q3**
> > >
> > > >**Response Q1.**
> > >
> > > Thank you for your timely feedback. First, we report the results for BC-safe in the following table across all the same simulations (cost limit $ l = 10 $). While BC-safe can guarantee a safe policy for some tasks, it struggles to learn a safe policy as task complexity increases. For instance, "CarButton2" is more complex than "CarButton1," as well as the CarGoal and CarPush tasks. For CDT, we were unable to complete all the simulations due to time constraints, as it requires significant time to train. Here, we provide its performance on the task "CarButton1" (cost limit  $ l = 10  $) and will include the results of CDT in our final revision. However, it is worth noting that, as demonstrated in FISOR (Table 1), CDT (with a cost limit of 10, the same as our setting in Table 1) fails to learn a safe policy for most tasks. Again, we would like to emphasize that our main contribution focuses on cases where the behavior policy is suboptimal. As we showed in Table 2 of our paper, the results in Table 1 aim to demonstrate that BARS can achieve better performance even when the behavior policy is good enough.
> > >
> > > | **Task**      | **Reward** | **Cost** |
> > > |---------------|------------|----------|
> > > | CarButton1    | 0.006      | 0.49     |
> > > | CarButton2    | -0.03      | **1.1**  |
> > > | CarGoal1      | 0.15       | 0.82     |
> > > | CarGoal2      | 0.10       | **3.67** |
> > > | CarPush1      | 0.18       | 0.92     |
> > > | CarPush2      | 0.28       | **1.43** |
> > > | AntVel        | 0.88       | 0.4      |
> > > | HalfCheetah   | 0.83       | 0.04     |
> > > | SwimmerVel    | 0.48       | 0.7      |
> > >
> > > *Table 1: BC-safe Performance*
> > >
> > > | **Task**      | **Reward** | **Cost**   |
> > > |---------------|------------|------------|
> > > | CarButton1    | 0.26       | **8.326**  |
> > >
> > > *Table 2: CDT Performance*
> > >
> > > ---
> > >
> > > >**Response Q2.**
> > >
> > > Thank you for your question. The behavior regularization issue becomes more severe when guaranteeing feasibility under multiple constraints, as we discussed in the paper ("2.1 LIMITATIONS OF EXISTING SOLUTIONS" - line 144). We made this claim because it is mathematically valid; however, we appreciate the reviewer's suggestion. We will leave this as future work since there is no commonly used benchmark for multiple constraints in offline safe-RL yet. Additionally, we have revised our statement to make it more precise.
> > >
> > > ---
> > >
> > > >**Response Q3.**
> > >
> > > Thank you for your question. First, we would like to clarify our previous response: The component
> > >
> > > $
> > > \left[w_{\text{safe}} \cdot \|z - z_{\theta}(a_t, s, t)\|^2_2\right]
> > > $
> > >
> > > in Eq. (9) is applied exclusively in diffusion-based models. If we understand your question clearly, do you mean whether a weight formulation such as
> > >
> > > $w_{\text{safe}}(s, a) = \exp\left(\mu_1 A^{\pi}_r(s, a) \cdot 1\_{\{\check{Q}_c(s, a) \leq l\}}\right)$
> > >
> > > could be implemented to train the \( Q \)-value network in CPQ? However, we do not believe that using the same weighted loss as in their paper is a fair comparison, as algorithms are designed differently and our approach incorporates additional components to address existing issues in offline safe RL. Modifying the weighted loss alone in CPQ would still not resolve the behavior regularization issue, as discussed in our paper. To address the reviewer's concern, we replace the policy training in CPQ from $[1(Q_c(s,a)\le l)Q_r(s,a)]$ to $[1(Q_c(s,a)\le l)Q_r(s,a)+1(Q_c(s,a)> l)\frac{Q_r(s,a)}{Q_c(s,a)}]$ , and report the performance on CarButton1 (Table 1 setting) with data ratios of 1:1 and 1:3 (Table 2 setting):
> > >
> > > | **Task**                        | **Reward** | **Cost**  |
> > > |---------------------------------|------------|-----------|
> > > | CarButton1 (whole dataset)      | 0.37       | **38.8**  |
> > > | CarButton1 (ratio 1:1)          | 0.29       | **32.3**  |
> > > | CarButton1 (ratio 1:3)          | 0.50       | **41.8**  |
> > >
> > > *Table 3: Modified CPQ on CarButton1*
> > >
> > > ---

---

> > > ### Author Response · Authors · 2024-11-23
> > > **Response to Reviewer Yqdv---Answer New question--Q4**
> > >
> > > >**Response Q4.**
> > >
> > > For this question, we previously interpreted the reviewer's comment as a suggestion to make our contribution more explicit, particularly emphasizing that "the strength of BARS should be its ability to achieve safety under safety-critical situations," as mentioned by the reviewer. In response, we modified the sentence in the abstract (lines 24–26) and revised our stated contributions accordingly. Regarding Table 3, we are unsure what additional details the reviewer requires. The purpose of Table 3 is to demonstrate that BARS is more flexible under different cost limits, and we only compare it with FISOR since both are region-based methods. It is evident that a higher reward can be achieved by increasing the cost limit, which aligns with expectations. We believe that our advantages are already well-explained and supported by the results in Table 2, which vary the performance of the behavior policy. Could the reviewer clarify further what additional information or analysis is required?
> > >
> > > ---
> > >
> > > Finally, if our response resolves your concerns to a satisfactory level, we wonder if the reviewer could kindly consider raising the score of your evaluation. Certainly, we are more than happy to address any further questions that you may have during the discussion period. We thank the reviewer again for the helpful comments and suggestions for our work.

---

> > > > ### Comment · Reviewer_Yqdv · 2024-11-29
> > > > **Remaining concerns**
> > > >
> > > > I thank the authors for their reply and the reviewer LCMx for raising the request about the remaining concerns. I still have concerns about this work. Particularly,
> > > >
> > > > (1) Overclaim issues. The authors overclaim their contributions in the original manuscript and the rebuttal reply. For example, during the rebuttal, the authors first state that this paper is the first paper to address how behavior regularization constraints in safe RL can lead to infeasible solutions, particularly with multiple constraints. After I pointed out that they do not have any experiments related to multiple constraints, they restated that the formulation is mathematically valid. I am not convinced that this is a significant contribution, because I believe many offline safe RL works can extend to multiple-constraint settings **from mathematical formulation**. For example, why are works such as CDT, BCQ-Lag, and FISOR not able to extend to multiple constraints **in mathematical formulation**?
> > > >
> > > > (2) Similarity with previous work. This work is based on one previous ICLR paper FISOR  [1], from the perspectives of model selection (diffusion model), weighted loss training, and formulation (feasibility-related). Although the authors reclaim their contribution after reading their revised manuscript, I still believe that this work is incremental compared to previous work. In Table 1, the experiment results of BARS and FISOR also look similar.
> > > >
> > > > (3) Reproducibility issues. The authors state that the previous ICLR 2024 paper FISOR can not be reproduced for experiments in Table 1. **This is a very severe problem**, meaning that at least one of these two works (FISOR and BARS) does not meet the requirements for publication in the ICLR conference. To validate the claim of the authors, I suggest the authors provide the **full code** with a README file and data files of their work so that we reviewers, and the public and examine the results and compare them fairly. Based on my previous experiments with FISOR, and the comments in the FISOR repo (https://github.com/ZhengYinan-AIR/FISOR/issues/4), I do not see any significant issues for reproducibility.
> > > >
> > > > (4) Confusing experiment results. In Table 1, the authors update the experiments for BC-Safe. Surprisingly, it shows a similar performance to the proposed BARS method: a slightly higher reward score (0.297 compared to BARS’s 0.282), and a minor safety violation (1.052 compared to 1). Considering that BC-safe is a very naive baseline, with MLP model structure, and intuitive optimization idea (behavior cloning with safe demonstration), the improvement of BARS on offline safe RL tasks is not significant with all their complex diffusion model and optimization schemes.
> > > >
> > > > Based on these remained concerns, I still recommend for **rejection** of this work.
> > > >
> > > > Reference:
> > > >
> > > > [1] Zheng, Yinan, Jianxiong Li, Dongjie Yu, Yujie Yang, Shengbo Eben Li, Xianyuan Zhan, and Jingjing Liu. "Safe offline reinforcement learning with feasibility-guided diffusion model." arXiv preprint arXiv:2401.10700 (2024).

---

### Official Review · Reviewer_1iK9 · 2024-11-02

**Soundness:** 3
**Presentation:** 3
**Contribution:** 2
**Rating:** 5
**Confidence:** 4

**Summary:**

This paper introduces Behavior-Aware Region-Based Safe offline RL (BARS), an algorithm designed to enhance safe offline RL by distinguishing between safe and unsafe states and applying region-specific behavior regularization. BARS adjusts its approach based on the safety level of each state, maximizing rewards in safe regions and minimizing costs in unsafe ones. BARS is theoretically demonstrated to achieve higher rewards while satisfying safety constraints, and experimental results on standard benchmarks show its superiority over baselines in both reward and safety performance.

**Strengths:**

1. The motivation behind BARS is clear and relevant to safe offline RL, particularly in optimizing policies while ensuring safety.
2. The design of BARS appears reasonable, as it introduces a novel approach to behavior regularization that adapts to the safety level of each state.
3. Evaluation results indicate that BARS significantly improves rewards and reduces costs compared to state-of-the-art methods, especially as dataset quality varies.

**Weaknesses:**

The primary concern is that BARS seems to achieve safety by over-conservatism. By learning value functions under the "most conservative policy" to identify safe and unsafe regions, BARS could limit its performance. In general, such a conservative policy can be the policy with minimal action, e.g., an agent choosing not to move at all to stay in a safe state to avoid constraint violation. Additionally, the evaluation results in Section 5 seem to show that BARS achieves both low costs and low rewards, e.g., the average normalized reward is 0.282 in Table 1 and 0.195 in Table 2.

**Questions:**

1. Following the weakness, could the authors compare their method with 1) the top 20 safe trajectories in the dataset that satisfy the cost limit, i.e., 10, and 2) Behavior cloning that only uses the safe trajectories in the dataset (BC-Safe in DSRL) for a more clear comparison?
2. The authors use IQL to learn the value function and Q-value function of the most conservative policy (Eq.(6), and (7)). However, according to Eq.(6) and (7), the states and actions are simply sampled from the whole dataset $\mathbb{E}_{(s, a) \sim \mathcal{D}}$. Could the authors clarify how this update can result in the value functions of the most conservative policy? The learned functions appear to represent those of the behavior policy used to collect the dataset.
3. Could the authors clarify the purpose of introducing an additional threshold, $\delta$, in Eq.(8)? $\delta$ seems to act as another cost threshold, similar to $l$ in Eq.(4) and (5). How is this $\delta$ selected, and is it related to the cost threshold $l$?
4. The authors use standard bellman loss to learn the $Q$ functions of the current policy (Eq.(14) and (15)). However, directly applying this technique in the offline RL setting is known to suffer from OOD actions. Why are OOD actions not an issue for BARS?
5. Regarding the experiment involving new data collection (lines 470-483), could the authors provide more details to assess if the proposed method is deemed safe enough for exploration (line 475)? For example, how many trajectories are collected in each environment, and what is the cumulative cost, i.e., total costs, incurred during this process?

---

> ### Author Response · Authors · 2024-11-21
> **Response to Reviewer 1ik9**
>
> We thank the reviewer for the constructive comments and feedback on our paper. We provide the detailed responses separately as follows:
>
> >**Response to lower costs and lower rewards:**
>
> We would like to emphasize that in safe RL, ensuring a safe policy (one with a cost lower than the threshold) is always the top priority. This is also a common observation in the offline safe RL benchmark (Liu et al., 2023a). We believe one reason for this is that the cost functions designed for different environments vary significantly, and balancing the reward and cost functions often requires environment-dependent parameters and careful fine-tuning. In our approach, since BARS uses a consistent set of hyperparameters across all environments in Table 2, we do not perform environment-specific fine-tuning. However, we can ensure a safe policy across all simulations. We believe that developing an adaptive hyperparameter selection method for different environments is an interesting direction for future work.
>
> ---
>
> >**Response to Q1:**
>
> Thanks for your suggestion. We implemented the BC-SAFE using the same dataset of CarButton1 we used in Table 2 to complete a comparison (Table [BC-SAFE Comparison](#tab-bcsafe-carbutton)). The result shows that BC-SAFE also fails to learn a safe policy.
>
> #### Table: BC-SAFE Comparison
> | Task          | Data Ratio | **Normalized Reward** | **Normalized Cost** |
> |---------------|------------|-----------------------|---------------------|
> | CarButton1    | 1:1        | 0.023                | 1.47                |
> | CarButton1    | 1:3        | 0.004                | 1.24                |
>
>
> ---
>
> >**Response to Q2:**
>
> In offline RL, the dataset is collected by some behavior policy. Thus, the optimization problem is to maximize the policy (for both RL and safe-RL) using only the offline dataset. The expectation is therefore taken over the behavior policy, which can be written as $ (s, a, s') \sim \mathcal{D} $. This is the reason why the distribution shift issue exists in offline RL—because the distribution of the dataset is different from the distribution under the policy we aim to optimize. We refer the reviewer to a more detailed explanation provided in this paper [R1].
>
> [R1] Levine, Sergey, et al. "Offline reinforcement learning: Tutorial, review, and perspectives on open problems." arXiv preprint arXiv:2005.01643 (2020).
>
> ---
>
> >**Response to Q3:**
>
> The value of the threshold $ \delta $ is the same as the cost limit $ l $ in our simulation. We define it as $ \delta $ in a more general form to control the robustness of the algorithm, as the $ Q $-functions may not be accurate and this is likely to hold true in practice.
>
> ---
>
> >**Response to OOD issue:**
>
> The OOD actions can be controlled by adding behavior regularization in the safe region. In the unsafe region, we directly minimize the cost function, which is trained using IQL. We thank the reviewer for raising this question, as it opens the possibility of further improvements by incorporating a penalty term into the critic network (as CPQ) to better control OOD actions.
>
> ---
>
> >**Response to Q5:**
>
> In the following table, we report the average cost per episode (cost limit $ l = 10 $) based on $ 600,000 $ data samples with 600 trajectories for the original offline dataset, the first collection (after training for $ 200K $ steps), and the second collection (after training for $ 400K $ steps). We can clearly observe that the mean episode cost drops dramatically after each new collection, which implies that the policy is safe enough for exploration.
>
> #### Table: Average Episode Cost
> | Dataset               | **Mean Episode Cost** |
> |-----------------------|-----------------------|
> | Offline Dataset       | 69.72                |
> | First collect data    | 7.45                 |
> | Second collect data   | 4.41                 |

---

> > ### Comment · Reviewer_1iK9 · 2024-11-23
> > **Additional clarifications are needed**
> >
> > I appreciate the authors' response and clarifications. However, my concerns and questions remain unresolved:
> >
> > Q1: The limited results of BC-Safe (evaluated in only one environment) are not sufficient to support the claim that "BC-Safe also fails to learn a safe policy." Additional evaluation in more environments is needed to substantiate this statement.
> >
> > Q2: The authors did not address my question. My question was not about distribution shifts. Let me rephrase: In lines 251–252, the authors state that "$\hat{Q}_c$ and $\hat{V}_c$ represent the cost Q-value function under the most conservative policy." However, in Eq.(6) and (7), the authors follow the IQL framework, which learns the Q-values of the behavior policy. My question is: How can the Q-values of the so-called most conservative policy be obtained by updating Eq.(6) and (7)?
> >
> > Q4: My concern about out-of-distribution (OOD) actions remains unaddressed. It is still unclear how OOD actions are controlled by adding behavior regularization in the safe region.
> >
> > Q5: The authors did not answer my question about the cumulative cost (i.e., total costs) incurred during the process. In online data collection, sample efficiency is crucial, especially in safe RL. Could the authors provide details on the cumulative cost incurred during this process?
> >
> > Additional question: the authors emphasized several times the advantage of their method regarding multiple constraints (e.g., in the abstract and contributions). However, based on the formulations and experiments, the method appears to address only a single constraint. It is unclear if the proposed approach truly has an advantage when dealing with multiple constraints. Could the authors clarify this?

---

> > > ### Author Response · Authors · 2024-11-24
> > > **Response to 1ik9---New response**
> > >
> > > >**Q1:**
> > >
> > > Thank you for your timely feedback. We report the results for BC-safe in the following table across all the same simulations (cost limit $ l = 10 $) and revised the experiment in Table 1. While BC-safe can guarantee a safe policy for some tasks, it struggles to learn a safe policy as task complexity increases. For instance, "CarButton2" is more complex than "CarButton1," as well as the CarGoal and CarPush tasks. We will include more simulations in our final version.
> > >
> > > | **Task**      | **Reward** | **Cost**   |
> > > |---------------|------------|------------|
> > > | CarButton1    | 0.006      | 0.49       |
> > > | CarButton2    | -0.03      | **1.1**    |
> > > | CarGoal1      | 0.15       | 0.82       |
> > > | CarGoal2      | 0.10       | **3.67**   |
> > > | CarPush1      | 0.18       | 0.92       |
> > > | CarPush2      | 0.28       | **1.43**   |
> > > | AntVel        | 0.88       | 0.4        |
> > > | HalfCheetah   | 0.83       | 0.04       |
> > > | SwimmerVel    | 0.48       | 0.7        |
> > >
> > > *Table 1: BC-safe Performance*
> > >
> > > ---
> > >
> > > >**Q2:**
> > >
> > > Thank you for clarifying the question. In Eq. (6) and (7), our aim is to learn the largest feasible/safe region using an offline dataset, which allows us to accurately identify this region (see Figure 1 in our paper). This largest feasible region implies that: (1) for states within this region, there exists at least one policy that can reach the destination without any violations; (2) for states outside of this region, no policy can guarantee a safe arrival at the destination. This is why we state that it 'represents the cost Q-value function under the most conservative policy.' Additionally, we provide a toy experiment where the agent starts in an unsafe region (i.e., outside the largest feasible region) in Appendix D.2.
> > >
> > > ---
> > >
> > > >**Q3:**
> > >
> > > In the safe region, we use the KL-divergence between our current policy and the behavior policy from the offline dataset as a constraint to avoid out-of-distribution (OOD) actions, specifically the term $ \|z - z_{\theta}(\alpha_t, s, t)\|^2_2 $ in Eq. (9). In the unsafe region, we use IQL to directly optimize the policy. The reason is that IQL is specifically designed to avoid OOD actions by using the state-conditional upper expectile of the action-value function $ Q(s, a) $ to estimate the value function $ V(s) $, thereby avoiding direct queries to the $ Q $-function with unseen actions. It serves as a strong baseline for offline RL by learning the value function using only dataset actions through quantile regression. While several variations of IQL have been proposed and could potentially be incorporated into our algorithm, we leave this exploration as future work.
> > >
> > > >**Q4:**
> > >
> > > In the following table, we report the average cost per episode (cost limit $l = 10$) based on 600,000 data samples with 600 trajectories for the original offline dataset, the first collection (after training for 200K steps), and the second collection (after training for 400K steps). We can clearly observe that the mean episode cost and **total cost (summation of all costs over the dataset)** drop dramatically after each new collection, implying that the policy is **safe enough** for exploration.
> > >
> > > |                         | **Mean Episode Cost** | **Total Cost** |
> > > |-------------------------|-----------------------|----------------|
> > > | Offline Dataset         | 69.72                 | 33182          |
> > > | First collected data    | 7.45                  | 4216           |
> > > | Second collected data   | 4.41                  | 3724           |
> > >
> > > *Table 1: Episode and Total Costs Comparison*

---

### Official Review · Reviewer_m36N · 2024-11-03

**Soundness:** 1
**Presentation:** 3
**Contribution:** 2
**Rating:** 3
**Confidence:** 4

**Summary:**

This paper studies behavior regularization in offline safe reinforcement learning. This paper argues that in unsafe regions where no safe policy exists, the optimized policy should not consider behavior regularization anymore. This is because (1) Enforcing safety constraints alongside behavior regularization often leads to conflicting objectives, making it hard to minimize the cost in unsafe regions; (2) Behavior policy can be very sub-optimal, strictly enforcing behavior regularization can be sub-optimal in this sense. To address these challenges, this paper proposes BARS (behavior-aware region-based safe offline RL), which firstly identifies safe and unsafe regions through expectile regression, and then discards the behavior regularizations for unsafe regions. The experiments on the DSRL benchmark demonstrate the effectiveness of the proposed method.

**Strengths:**

1. This paper is well-written, presenting a clear motivation for the problem setups.

2. Behavior-aware regularization is necessary for safe offline RL.

**Weaknesses:**

## The motivation is good but some method details are quite strange
1. Eq. (7) tries to ensure $(1-\alpha) c + \alpha(c + V_c)=Q_c$. This is not a common Bellman equation for the cost value Q and V function. Instead, this equation is similar to the one for the feasible value function but is still different: $(1-\alpha) h + \alpha \max (h, V_h) =Q_h$. Specifically, for the feasible value function, a maximization term exists, but in Eq. (7), the authors directly replace it as a summation term.

2. The definition of $A_r^\pi$ in Eq. (9) is somehow ad-hoc, solely providing some intuitive explanations.

3. It is strange that directly minimizing the $Q_c$ value in Eq.(11) will not lead to a large bootstrapping error accumulation.

4. If not, this means that the policy under unsafe regions still stay near the behavior policy. So the introduction of TRPO-style optimization in Eq. (12) still tries to ensure a relatively relaxed behavior regularization, which contradicts the motivation of this paper that the behavior regularization for unsafe regions should be dropped.

5. In my view, the potential benefits of BARS are primarily attributed to the definition of $A_r^\pi$. Under its definition, the policy will exhibit a more conservative behavior to avoid unsafe regions and so is safer. For unsafe regions, it is quite hard for me to judge if the policy can obtain a reasonable behavior since no behavior regularization exists anymore and the policy can easily exploit the approximation errors of the Q_c value function. It would be better if the authors could show more rollout trajectories for BARS in Figure 1 when starting at one unsafe region.

6. The authors have identified the safe and unsafe regions through expectile regression, but still need to learn additional Q and Qc value functions through standard bellman update in Eq.(14-15). This can be unstable due to error accumulations and inefficient due to the costly diffusion sampling process.

## Evaluations
7. Table 3 shows that FISOR produces different results for varied cost limits. However, FISOR is one cost limits-agnositc method that studies hard constraint and should not behave differently with different cost limits.

**Questions:**

Please see Weakness for details.

---

> ### Author Response · Authors · 2024-11-21
> **Response to Reviewer m36N---Part 1**
>
> We thank the reviewer for the constructive comments and feedback on our paper. We provide the detailed responses separately as follows:
>
> >**Response to Bellman equation:**
>
> In the paper, we identified a typo in the Bellman equation (we missed the discount factor). The correct form is:
> $$
> \mathcal{L}_{\check{Q}_c}= \mathbb{E}\_{(s,a,s') \sim \mathcal{D}} \left[ \left( (1-\alpha)c(s,a) + \alpha(c(s,a) + \gamma \check{V}_c(s')) - \check{Q}_c(s,a) \right)^2 \right]
> $$
>
> The term $ (1-\alpha) $ is included to model the scenario where, with probability $ 1 - \alpha $, an episode ends after the current time step (e.g., after taking a very unsafe action). It is easy to show that $ c(s,a) + \gamma \check{V}_c(s') = \check{Q}_c(s,a) $ as $ \alpha \rightarrow 1 $ (we use $ \alpha = 0.99 $ in our simulation). Importantly, FISOR defines the feasible region using the value function for the hard constraint based on Hamilton-Jacobi (HJ) reachability from control theory (hence the inclusion of the maximization term). However, this formulation applies only to **deterministic systems**, which are rarely encountered in most RL environments, and cannot be generalized to problems with more flexible cost limits. As we indicated in our paper, the Bellman equation used in FISOR to train the value function does not hold in stochastic systems.
>
> ---
>
> >**Response to definition of $A^{\pi}_r$:**
>
> We would like to emphasize that the definition of $ A_r^\pi $ is not ad hoc; it directly follows from our theoretical formulation with practical reasoning. Due to the difficulty in obtaining accurate estimates of $ Q^\pi_r $ and $ Q^\pi_c $ (which is likely to be true in practice due to the use of neural networks), we implemented a practical approach to penalize the reward $ Q $-function in $\text{Eq.(8)}$. Specifically, we penalize the reward $ Q_r^\pi $-function based on the cost $ Q_c^\pi $-value under the current policy whenever $ Q_c^\pi(s, a) > \delta $, where $ \delta $ is the cost limit (other parameters can also be used to control the robustness of the algorithm). This indicates that the current policy is not safe. When the policy is safe, we directly optimize $ Q_r^\pi $. Given that the estimation may not be accurate for both $ Q $-functions, we penalize the reward $ Q $-function according to the magnitude of the cost $ Q $-function, assuming that the trend of the $ Q $-values is well learned, even if the estimation is not precise.
>
> ---
>
> >**Response to bootstrapping error accumulation:**
>
> As described in our algorithm, it is divided into two distinct stages. First, we learn the largest safe region, which is determined by the **most conservative** policy. Specifically, this ensures that for any state within this region, there exists no unsafe policy. Using IQL to learn the value function and achieve $ \check{V}_c(s) = \min_a \check{Q}_c(s, a) $ under the most conservative policy is both straightforward and efficient. This approach can be implemented using offline data only, avoiding the need to train the policy network. As a result, it provides an accurate estimation, is time-efficient, and eliminates bootstrapping error issues.
>
> We believe the reviewer may have misunderstood the TRPO used in $\text{Eq.(12)}$. In our algorithm, we introduce the parameter $ K_{\text{trust}}$, which indicates the point after which we optimize the policy using $\text{Eq.(11)}$ and $\text{Eq.(12)}$. This ensures that the current policy does not diverge significantly from the previous one by considering the trust region. This occurs **only** when the policy is safe enough. **Besides, we would like to emphasize whether only using $\text{Eq.(11)}$ can still ensure learning a safe policy (as shown in Table 4).**
>
> In our simulation, we simply set $ K_{\text{trust}}$ to half of the total training episodes without any tuning. Additionally, we would like to clarify that in $\text{Eq.(12)}$, we do not use behavior regularization from the offline dataset; instead, we employ the previously trained policy (target policy) for regularization. The parameter $ k $ specifies how often we update the target policy, analogous to the target network updates used in DQN. We have clarified these points in our revision. This approach does not violate our formulation.
>
> ---

---

> > ### Author Response · Authors · 2024-11-21
> > **Response to Reviewer m36N---Part 2**
> >
> > >**Response to weakness 5:**
> >
> > Our practical version includes several key components: a safe-region-based approach, a different $ A_r^\pi $ for controlling the uncertainty and inaccurate estimation of $ Q $-functions, and a gradually updated target policy for controlling the trust region. To further address the reviewer's concern, we have included a figure in Appendix D.2 that demonstrates the policy starting from unsafe regions.
> >
> > ---
> >
> > >**Response to Weakness 6:**
> >
> > The training of additional $ Q $-functions, $ Q_c^\pi $ and $ Q_r^\pi $, is necessary for optimizing the policy within the safe region, as the objective is to maximize the policy while maintaining safety. This is a common approach in actor-critic methods, where techniques like double networks and truncated functions are often used to stabilize the $ Q $-functions. We would be happy to discuss this further if the reviewer has any additional suggestions.
> >
> > The training time for our algorithm is comparable to existing approaches. We have included a table in the appendix to show the training times, and we can observe that the results are consistent. We do believe BARS is not time-consuming, and our reported results are averaged over different random seeds, which shows that the results are quite stable.
> >
> > ---
> >
> > >**Response to Weakness 7:**
> >
> > FISOR is designed to be cost limit-agnostic due to its reliance on a hard constraint framework. However, as we indicated in our paper, this design limits the generalization of their approach to other scenarios where a specific cost limit is considered, which is a more practical setting. Our aim is to demonstrate that increasing the cost limit should potentially lead to better performance for an algorithm.
> >
> > We do not understand why the reviewer considers this to be a weakness of our paper, especially since FISOR is one of the most closely related works and it is reasonable to add more discussion.

---

> > ### Comment · Reviewer_m36N · 2024-11-22
> > **Thanks for the responses**
> >
> > Thanks for the effort and responses. However, my concerns are not well-resolved.
> >
> > ## Bellman Equation
> > I understand that FISOR only applies to deterministic systems. However, I am still confused about why do not authors directly use the standard Bellman Equation in IQL to learn the largest feasible region but adopt a stochastic version from FISOR. In my view, FISOR uses that Bellman equation to approximate a fixed point iteration when $\gamma\rightarrow 1$ since the original definition of $Q_h^*$ in FISOR is not a contraction mapping. However, it is unnecessary to adopt that design if the author considers the stochastic environment. The authors can simply use $r +\gamma V = Q$. So, it would be good if the authors could explain this.
> >
> > ## Bootstapping error
> > I am curious why directly minimizing the cost function in unsafe regions (Eq. (10)) will not lead to error accumulation caused by OOD actions. However, the authors did not fully address this concern. I noticed that the reviewer 1iK9 also pointed out this issue.
> >
> > ## Weakness 7
> > FISOR is one cost limit-agnostic method, so its performance should remain consistent across different cost limits in Table 3. However, it seems that the authors got different behaviors of FISOR given different cost limits. For example, FISOR deteriorates when the cost limits are relaxed from 10 to 40, which is strange.

---

> > > ### Author Response · Authors · 2024-11-23
> > > **Response to Reviewer m36N---New response**
> > >
> > > >**Bellman Equation**
> > >
> > > The use of $ 1-\alpha $ represents the probability that an episode may terminate after this step with the probability $1-\alpha $. We adapt the Discounted Reach-Avoid Bellman Equation [R1, R2] to the standard Bellman Equation setting for stochastic environments. We believe that the formulation with an absorbing probability is more practical. While we do not observe significant differences between using Eq. (6) and the standard Bellman equation in our simulations (as we set $\alpha = 0.99 $), we think this approach has potential applications in environments where episodes can terminate early due to absorbing states.
> > >
> > > [R1] Hsu, Kai-Chieh, Vicenç Rubies-Royo, Claire J. Tomlin, and Jaime F. Fisac. "Safety and liveness guarantees through reach-avoid reinforcement learning." arXiv preprint arXiv:2112.12288 (2021).
> > >
> > > [R2] So, Oswin, Cheng Ge, and Chuchu Fan. "Solving Minimum-Cost Reach Avoid using Reinforcement Learning." arXiv preprint arXiv:2410.22600 (2024).
> > >
> > > ---
> > >
> > > >**Bootstapping error**
> > >
> > > In the unsafe region, we use IQL to directly optimize the policy. The reason is that IQL is specifically designed to avoid OOD actions by using the state-conditional upper expectile of the action-value function $Q(s, a) $ to estimate the value function $ V(s) $, thereby avoiding direct queries to the $ Q $-function with unseen actions. It serves as a strong baseline for offline RL by learning the value function using only dataset actions through quantile regression. While several variations of IQL have been proposed and could potentially be incorporated into our algorithm, we leave this exploration as future work.
> > >
> > > ---
> > >
> > > >**Weakness 7**
> > >
> > > Thank you for your question. For Table 3, we conducted additional simulations for FISOR and averaged the results over three random seeds. Since FISOR is a cost limit-agnostic method, comparing performance across different cost limits is unnecessary. Thus, we modified Table 3 in our revised paper. However, the purpose of Table 3 is to demonstrate that BARS is more adaptable to practical scenarios where varying cost limits are required.

---

> > > ### Comment · Reviewer_LCMx · 2024-11-29
> > >
> > > Hi, Reviewer m36N
> > >
> > > I would like to know whether the authors have addressed your concerns. I'm looking forward to your reply.

---

> ### Comment · Reviewer_m36N · 2024-12-02
>
> Thanks for the responses. However, the concerns are still unresolved.
>
> 1. **Bellman Equation**. The authors argure that they adopt the Discounted Reach-Avoid Bellman Equation, which is $\mathbb{B}(Q^*(s,a))= (1-\gamma) h(s) + \gamma (\max (h(s), V^*(s)))$.  Note that this original equation has a `max` term, so the $(1-\gamma) h(s) + \gamma (\max (h(s), V^*(s)))$ will not degenerate to $h(s)+\gamma V^*(s)$. However, the authors just naively adopt this equation, where the $-\alpha c$ and $\alpha c$ in Eq. (6) will cancel out, making Eq. (6) just $c+\gamma \times \alpha V$, which seems that the carefully designed $\alpha$ term just adjust the discounting factor, rather than being the termination probability argued by the authors. I noticed the `reviewer RHPd` also pointed out this concern.
>
> 2. **Bootstrapping Error**. I disagree with the author that just adopting expectile regression to learn the value function while not regularizing the policy can avoid OOD issues. The value function in IQL is learned through in-sample learning, which can avoid bootstrapping, but still has some overestimated OOD regions. The value is stable since the value learning and policy learning is decoupled, and the value function will never querry these OOD regions. However, these OOD regions EXIST. So, during policy optimization, IQL still requires behavior regularization (`AWR`, which equivalents to a implicit policy regularization w.r.t a KL-divergence constriant and the objective is $\min_\theta - \mathbb{E}_{(s,a)\sim\mathcal{D}}(\log \pi(a|s)exp(A)$. Note all data are sampled from the dataset $\mathcal{D}$) to prevent exploiting on these OOD regions.  However, according to Eq. (10), the authors try to directly optimize the cost value function **without any regularization**, and the objective is $\min\mathbb{E}\_{s\sim\mathcal{D}, a\sim\pi} exp(A)$. `Note that here, the action $a$ is sampled from the policy $\pi$, and this objective has NO regularization.` So, I can tell that the policy optimization obejective is not IQL as the author said: `we use IQL to directly optimize the policy`.**I implemented a lot of offline RL methods and their variants, but found no one can obtain stable results without any behavior regularization when directly maximizing the Q value obtained by IQL**.
>
> 3. **Weakness 7**. Yes, FISOR is one cost-limit agnostic method. But, how can the authors get totally different results (the costs are varied from 0.06 to 0.36) for different cost limits in their first submission. After I pointed this out, the authors directly removed this result, without comparing to any baselines. There are some safe offline RL methods that can handle varied cost limits, so why did not the authors compare against these baselines? Also, I noticed that the FISOR results reported by the authors are not consistent with the original FISOR paper, which is also noticed by the `reviewer Yqdv`. I found one recent ICLR submission (Constraint-Conditioned Actor-Critic for Offline Safe Reinforcement Learning) that also reproduced FISOR and reported far more better results, even compared to the original ones. So, this paper potentially not well reproduced the baselines and the experimental results are not convincing.
>
> So, regretfully, I cannot raise my score and recommend **rejection** for this paper.

---

### Official Review · Reviewer_oNEJ · 2024-11-03

**Soundness:** 3
**Presentation:** 2
**Contribution:** 2
**Rating:** 5
**Confidence:** 3

**Summary:**

The paper introduces BARS (Behavior-Aware Region-Based Safe offline RL), which differentiates between safe and unsafe states and applies selective behavior regularization tailored to each state’s safety level. BARS prioritizes reward maximization and adhers to safety constraints in safe states while minimizing risks in unsafe ones. Experimental results show that BARS outperforms existing methods, achieving higher rewards and better constraint satisfaction across varying data quality levels.

**Strengths:**

- The paper is supported by both theoretical and experimental results.
- Solving different optimization problems for safe states and unsafe states sounds effective to me in the offline safe RL setting.
- The issues of infeasible solutions and poor data quality identified in the paper are important.
- The experimental setting in Figure 1 (safe offline RL with new data collection) appears to have high practical value.
- The empircal performance of BARS appears good.

**Weaknesses:**

- I am not quite certain about the contributions of this work compared to FISOR, especially before noticing Appendix B. Since BARS and FISOR are closely related, it would enhance the clarity of the paper to include this discussion in the main text, if space permits. Additionally, pointing out which component or improvement of BARS, in comparison to FISOR, contributes the most to its performance would further clarify the contributions of the paper.
- While the gap between theoretical and practical is inevitable, the authors are encouraged to elaborate on the differences between the practical algorithm and the theoretically analyzed algorithm (Eq. (5)).

[1] Zheng et al. Safe Offline Reinforcement Learning with Feasibility-Guided Diffusion Model.

**Questions:**

- Table 4 shows that policy restriction increases reward at the risk of higher cost. It may be difficult to determine which one is better. How do the authors recommend balancing the increased reward against the higher cost when using policy restriction?

---

> ### Author Response · Authors · 2024-11-21
> **Response to Reviewer oNEJ---Part1**
>
> We thank the reviewer for the constructive comments and feedback on our paper. We provide the detailed responses separately as follows:
>
> >**Response to different as FISOR**
>
> Our work focuses on the existing issue of using behavior regularization in offline safe RL. FISOR and all the other baselines cannot ensure a safe policy when the behavior policy is suboptimal. Please see our detailed response in **[General Response](https://openreview.net/forum?id=GVhfWu5L8D&noteId=wLbYJJ4AuL)**.
>
> ---
>
> >**Response to "pointing out which component or improvement of BARS"**
>
> We claim that the new optimization framework we proposed in $\text{Eq.(5)}$ is the key component in mitigating regularization issues. To further validate our approach, we conducted an experiment (Table [Component Analysis](##table-component-analysis)) in the CarButton2 environment (data ratio 1:1 in Table 2) to demonstrate that our key components are essential for learning a safe policy under different data distributions.
>
> In this experiment, we divide the following four situations and report the results in Table [Component Analysis](##table-component-analysis):
>
> 1. (i) $ {D}(\pi||\pi_{\beta})\le\epsilon $ only in the safe region.
> 2. (ii) $ {D}(\pi||\pi_{\beta})\le\epsilon $ both in safe and unsafe regions.
> 3. (a) $ \frac{Q_r^\pi(s,a)}{Q_c^\pi(s,a)} \cdot 1_{\{Q_c^\pi(s,a) > \delta \}} + Q_r^\pi(s,a) \cdot 1_{\{Q_c^\pi(s,a) \leq \delta \}} $.
> 4. (b) $ {Q_r^\pi(s,a)} \cdot 1_{\{Q_c^\pi(s,a) > \delta \}} + Q_r^\pi(s,a) \cdot 1_{\{Q_c^\pi(s,a) \leq \delta \}} $.
>
> This experiment shows that using (ii) cannot assure a safe policy, and component (i) is the crucial factor to guarantee a safe policy. The results highlight how each component of BARS contributes to learning a safer policy.
>
> #### Table: Component Analysis
> | **Component**        | **Normalized Cost** | **Normalized Reward** |
> |-----------------------|---------------------|-----------------------|
> | Using (ii), (b)        | 2.69               | -0.05                |
> | Using (i), (b)        | 0.785              | -0.07                |
> | Using (ii), (a)        | 1.43               | -0.01                |
> | Using (i), (a) (**BARS**) | 0.63               | -0.08                |
>
> ---
>
> >**Response to "summary of contributions" and limitations**
>
> Thanks for the suggestion, we revised the contribution in the revised draft. Here, we will provide a brief summary of our contributions.
>
> 1. We propose a new optimization framework that rethinks the need for behavior regularization in offline safe reinforcement learning. First, we determine safe and unsafe regions using the correct formulation of the Bellman equation, then remove behavior regularization in unsafe regions. This framework helps the agent learn safe actions across various data distributions, even when some datasets are of low quality.
> 2. We introduce a cautious updating method called "Region-Based Risk-Reward Optimization," which enables the agent to avoid unsafe actions, even when starting in a safe region.
>
> ---

---

> > ### Author Response · Authors · 2024-11-21
> > **Response to Reviewer oNEJ---Part2**
> >
> > >**Response to "gap between theoretical and practical"**
> >
> > We follow exactly the same approach for controlling behavior regularization. The practical version of our approach has the following differences: 1. Due to the difficulty in obtaining accurate estimates of $ Q^\pi_r  $ and  $ Q^\pi_c  $ (which is likely to be true in practice due to the use of neural networks), we implemented a practical approach to penalize the reward  $ Q  $-function in Eq.(8). Specifically, we penalize the reward  $ Q_r^\pi $-function based on the cost $ Q_c^\pi  $-value under the current policy whenever  $ Q_c^\pi(s, a) > \delta  $, where  $ \delta  $ is the cost limit (we can also use other parameters to control the robustness of the algorithm). This indicates that the current policy is not safe. When the policy is safe, we directly optimize  $ Q_r^\pi $. Given that the estimation may not be accurate for both  $ Q  $-functions, we penalize the reward  $ Q  $-function according to the magnitude of the cost  $ Q  $-function, assuming that the trend of the  $ Q  $-values is well learned, even if the estimation is not precise. 2. We use a parameter  $ K_{\text{trust}}  $, which indicates the point after which we optimize the policy using $\text{Eq.(11)}$ and $\text{Eq.(12)}$. This ensures that the current policy does not diverge significantly from the previous one by considering the trust region. This occurs  ${\bf only} $ when the policy is safe enough. In our simulation, we simply set  $ K_{\text{trust}}  $ as half of the total training episodes without any tuning. Additionally, we would like to clarify that in $\text{Eq.(12)}$, we do not use behavior regularization from the offline dataset; instead, we employ the previously trained policy (target policy) for regularization. The parameter  $ k  $ specifies how often we update the target policy, analogous to the target network updates used in DQN. We have made these points clear in our revision.
> >
> > ---
> >
> > >**Response to higher rewards but higher costs**
> >
> > We would like to emphasize that in safe RL, ensuring a safe policy (one with a cost lower than the threshold) is always the top priority. This is also a common observation in the offline safe RL benchmark (Liu et al., 2023a). Balancing the reward and cost functions often requires environment-dependent parameters and careful fine-tuning. By applying the policy restriction in $\text{Eq.(12)}$, BARS improves its reward performance **while satisfying the constraint, even if the cost increases slightly**, thereby eliminating the need for careful parameter selection. **Besides, we would like to emphasize whether using $\text{Eq.(11)} $or $\text{Eq.(12)}$ can both ensure learning a safe policy (as shown in Table 4).**
> >
> > ---

---

### Official Review · Reviewer_LCMx · 2024-11-04

**Soundness:** 3
**Presentation:** 3
**Contribution:** 2
**Rating:** 6
**Confidence:** 4

**Summary:**

The paper proposes a novel algorithm, BARS, aimed at addressing the issue of behavior regularization in offline safe reinforcement learning. The authors effectively tackle the challenge of learning safe strategies in unsafe states by differentiating between safe and unsafe states and applying region-based selective behavior regularization to optimize the policy.

The authors start from **two main drawbacks** of prior offline safe RL methods: one is the importance of finding less harmful action, and the other is the imitation of behavior policy. The method is based on FISOR [1], a SOTA method in offline safe RL. It first finds the largest safe region and an absolute unsafe region by HJ-reachability. Then, it further separates the safe region into absolute safe and margin safe by using a threshold. A **main improvement** is removing the behavior constraint in the unsafe region to better learn a safe policy. The experiments involving different coverage of datasets demonstrate its effectiveness.

However, the experiment lacks analysis of hyperparameters and some derivations are confusing.

Reference:
[1] Zheng, Y., Li, J., Yu, D., Yang, Y., Li, S. E., Zhan, X., and Liu, J. (2024). Safe offline reinforcement learning with feasibility-guided diffusion model. arXiv preprint arXiv:2401.10700.

**Strengths:**

- The paper removes behavior cloning in unsafe regions, which is straightforward, effective and relatively novel.
- It solves the main weaknesses of some prior work and clearly explains how to improve previous methods.
- The authors improve the previous optimization problem and replace it with a new one. They also provide a theorem as to why it is better than the original one.
- The main experiments show the superiority of BARS and ablation studies provide interesting findings.
- The additional experiments (adding restrictions when updating the policy and different data collection) are interesting. It explores the performance under different behavior policies and extends it to more practical experiments.

**Weaknesses:**

- The overall method is based on FISOR, which lacks some novelty.
- Extensive experimental results indicate that BARS seems to struggle to balance the reward and the cost. While the main results in some tasks have lower costs with lower rewards (e.g. CarGoal1), the results in other tasks show higher rewards and costs (e.g. AntVelocity).
- The introduction section lacks a summary of contributions, which would help readers quickly understand the paper. (minor suggestion)
- I suspect the good performances are due to adjustment of hyperparameters (such as safe loss control).
- The paper lacks a Limitation section which is important.
- Some derivations of the formulas seem to be confusing. See the Question for details.

**Questions:**

1. In line 264, Why can IQL be used to obtain results $V_c(s) = \min_a Q_c(s,a)$?

2. In line 270, expectile regression in IQL is $L^{\tau}(u) = |\tau - 1(u<0)| u^2$. Why can it use $u>0$?

3. In line 300, why does $A^{\pi}_r(s,a)$ have this form? Why do not use the advantage function?

4. In line 318, if there are no distributional shifts, how can equation 11 be deviated? (I think there is no exp.)

5. Since the method introduces new hyperparameters (such as threshold $\delta$, trust episode $K_{trust}$), how do they affect the performances? The threshold $\delta$ is not contained in Table 7.

6. In the Algorithm 1, what's the meaning of $k$?

Recommendation:
Due to the absence of significant inherent errors in the method and some experimental settings are quite interesting, I would like to give 'borderline accept'.

---

> ### Author Response · Authors · 2024-11-21
> **Response to Reviewer LCMx---Part 1**
>
> We thank the reviewer for the constructive comments and positive feedback on our paper. We provide the detailed responses separately as follows:
>
> ---
>
> > **Difference between BARS and FISOR**
>
> Our work focuses on the existing issue of using behavior regularization in offline safe RL. FISOR and all other baselines cannot ensure a safe policy when the behavior policy is suboptimal. Please see our detailed response in **[General Response](https://openreview.net/forum?id=GVhfWu5L8D&noteId=wLbYJJ4AuL)**.
>
> ---
>
> >**Response to lower costs and lower rewards**
>
> We would like to emphasize that in safe RL, ensuring a safe policy (one with a cost lower than the threshold) is always the top priority. This is a common observation in the offline safe RL benchmark (Liu et al., 2023a).
>
> We believe one reason for this is that cost functions vary significantly across environments, and balancing reward and cost functions often requires environment-dependent parameters and careful fine-tuning. In our approach, BARS uses a consistent set of hyperparameters across all environments in Table 2 without environment-specific fine-tuning. However, we can ensure a safe policy across all simulations. Developing an adaptive hyperparameter selection method for different environments is an interesting direction for future work.
>
> ---
>
> >**Response to "summary of contributions" and limitations**
>
> Thank you for the suggestion. We have revised the contribution section in the introduction and added a limitations section in the revision.
>
> >**Summary of Contributions:**
> - We propose a new optimization framework that addresses the existing issues of using behavior regularization in offline safe reinforcement learning.
> - We introduce a practical algorithm, BARS, which builds on our theoretical findings and enables the agent to optimize the policy separately in different regions, determined by the feasible region.
> - We determine safe and unsafe regions using the correct formulation of the Bellman equation, then remove behavior regularization in unsafe regions. This framework helps the agent learn safe actions across various data distributions, even when some datasets are of low quality.
>
> ---
>
> >**Response to Hyperparameters**
>
> We emphasize that in our experiments, we did not perform a hyperparameter search. Instead, we used a single set of loss control parameters, $b_1 = 1$ and $b_2 = 3$, for all tasks in Table 2 and most tasks in Table 1. The exceptions were "AntVel" and "HalfCheetah" in Table 1, where we set $b_1 = 3$ and $b_2 = 1$ to achieve higher rewards. As demonstrated, **no** other baselines can ensure safety due to the regularization issue we highlighted. Please see our detailed response in **[General Response](https://openreview.net/forum?id=GVhfWu5L8D&noteId=wLbYJJ4AuL)**.
>
> ---
>
> >**Response to using IQL**
>
> As indicated in our paper, solving the optimization problem in $ \text{Eq.(4)} $ is challenging because:
> 1. The $ Q $-networks and policy are parameterized by neural networks, which introduces instability issues.
> 2. Accurate estimation is difficult, requiring a long training period even for evaluating a single policy.
>
> To address these, we solve an alternative optimization problem under the largest safe region, determined by the **most conservative** policy. This ensures that for any state within this region, no unsafe policy exists. Using IQL to learn the value function and achieve $\check{V}_c(s) = \min_a \check{Q}_c(s, a)$ under the most conservative policy is both straightforward and efficient. This method uses offline data only, avoiding policy network training, providing accurate estimation, being time-efficient, and eliminating instability.
>
> ---
>
> >**Response to $u$ in IQL**
>
> In our loss function, we prioritize $u > 0$ as it learns the value function under the most conservative policy. This is a minimization problem rather than the maximization problem used in the IQL paper. Therefore, the asymmetric loss function operates in the opposite direction. We have clarified this point in the revised paper.
>
> ---

---

> ### Author Response · Authors · 2024-11-21
> **Response to Reviewer LCMx---Part 2**
>
> >**Response to $A_r^\pi(s, a)$**
>
> We penalize the reward $Q_r^\pi$-function based on the cost $Q_c^\pi$-value whenever $Q_c^\pi(s, a) > \delta$ (where $\delta$ is the cost limit). This indicates that the policy is unsafe. When the policy is safe, we directly optimize $Q_r^\pi$.
>
> Given that $Q$-function estimation may not always be accurate, we penalize $Q_r^\pi$ based on the cost $Q_c^\pi$, assuming the $Q$-value trends are well-learned even if the estimation isn't precise. We do not include a neural network for $V_r^\pi$, as we consider it redundant, though it would work if trained. This choice avoids redundancy as both advantage and $Q$-networks have been extensively discussed in RL literature.
>
> ---
>
> >**Response to distributional shifts**
>
> If no distributional shifts exist, behavior regularization isn't required when learning the policy. This is only true when the behavior policy closely resembles the optimal policy. In $ \text{Eq.(11)} $, the regularization term is not required, as discussed in our optimization problem in $ \text{Eq.(5)} $.
>
> ---
>
> >**Response to $\delta$, $k$, and $K_{trust}$**
>
> - The threshold $\delta$ is the same as the cost limit $l$ in our simulation but is defined in a general form to control the algorithm's robustness since $Q$-functions may be inaccurate in practice.
> - $K_{\text{trust}}$ ensures the policy does not diverge significantly from the previous one. It is applied only when the policy is safe enough. In our simulation, \(K_{\text{trust}}\) is set as half the total training episodes without tuning.
> - $k$ specifies the update frequency for the target policy, similar to target network updates in DQN.
>
> Finally, we emphasize that **not using $\text{Eq.(12)}$** can still ensure learning a safe policy (as shown in Table 4).
>
> ---

---

> > ### Comment · Reviewer_LCMx · 2024-11-22
> >
> > Thank you for explaining. I fully understand why IQL has this form here and it seems reasonable. However, I still have some concerns as follows. I'd like to get some explanations beyond the paper.
> >
> > 1. I have known what $A_r^{\pi}(s,a)$ means. However, FISOR optimizes the advantage function here. Why do you use $\frac{Q_r^\{\pi}(s,a)}{Q_c^\{\pi}(s,a)}1(Q_c^{\pi}>\delta) + Q_r^{\pi}(s,a)1(Q_c^{\pi}(s,a)\leq\delta)$ instead of $\frac{A_r^\{\pi}(s,a)}{A_c^\{\pi}(s,a)}1(Q_c^{\pi}>\delta) + A_r^{\pi}(s,a)1(Q_c^{\pi}(s,a)\leq\delta)$. Here, $A(s,a)$ is the advantage function.
> >
> > 2. I can understand why there is no distribution shift. However, please give a detailed deviation of Eq.11 (from Eq.4 when the optimization problem does not require $D(\pi || \pi_{\beta}) \leq \epsilon$), especially explain how the weight $exp(-\mu_2 A_c(s,a))$ comes from.
> >
> > 3. I'd like to see some additional ablation experiments about $K_{trust}$. I am concerned that you may have used it when the policy was not safe enough.

---

> > > ### Author Response · Authors · 2024-11-25
> > > **Response to LCMx---New Response**
> > >
> > > >**Q1:**
> > >
> > > Thank you for your timely feedback. We do not train a neural network for $ V_r^\pi $; instead, we use a standard actor-critic network to learn the $ Q $-values. However, we believe that the advantage network could also work if implemented. Note that FISOR uses an advantage network because it employs IQL to learn the largest cumulative reward function while ignoring the cost and directly optimizing that. As discussed in our paper, this approach may have issues since the $ Q $-value under the greedy policy is not guaranteed to be safe and is unlikely to be safe in a safe RL setting since the $ Q $ (advantage) function is the $ Q $ function under the training policy.
> > >
> > > >**Q2:**
> > >
> > > For optimizing the value function obtained from $ IQL $, we first observe that:
> > >
> > > $$
> > > \max\_{\pi, a \sim \pi}  \mathbb{E} \left[ -\check{A}\_c(s, a) \right]
> > > \overset{(i)}{\iff} \min\_{\pi, a \sim \pi}  \mathbb{E} \left[ \check{Q}\_c(s, a) \right]
> > > \overset{(ii)}{\iff} \min\_\pi  V^{\pi}\_c(s)
> > > \overset{(iii)}{\iff} \max\_\pi  -V^{\pi}\_c(s),
> > > $$
> > >
> > > where (i) follows from $ \check{A}\_c(s, a) = \check{Q}\_c(s, a) - \check{V}\_c(s) $ and the fact that $\check{V}\_c(s) $ is independent of actions. Step (ii) follows from the definitions, and step (iii) is a straightforward reversal.
> > >
> > > Next, we analyze the optimization in the unsafe region for Eq. (11):
> > >
> > > $$
> > > \max\_{\pi, a \sim \pi}  \mathbb{E} \left[ -\check{A}\_c(s, a) \right]
> > > $$
> > >
> > > $$
> > > \text{s.t.} \quad \int\_{a} \pi(\cdot \vert s) \, da = 1, \quad D\_{\text{KL}}(\pi || \pi') \leq \epsilon,
> > > $$
> > >
> > > where $ \pi' $ is a copy of the target policy network, not the behavior policy \( \pi_{\beta} \).
> > >
> > > The corresponding Lagrangian function can be written as:
> > >
> > > $$L(\pi, \lambda, \nu) = \mathbb{E} \left[ -\check{A}\_c(s, a) \right]- \lambda \left( \int\_{a} \pi(\cdot \vert s) \, da - 1 \right)-\nu \left( D\_{\text{KL}}(\pi || \pi') - \epsilon \right)$$
> > >
> > > Taking the derivative with respect to $ \pi $, we get:
> > >
> > > $$
> > > \frac{\partial L}{\partial \pi} = -\check{A}\_c(s, a) - \lambda - \nu \left( D\_{\text{KL}}(\pi || \pi') - \epsilon \right)
> > > = -\check{A}\_c(s, a) - \lambda - \nu (\log \pi - \log \pi') + \nu.
> > > $$
> > >
> > > Setting the derivative to zero yields:
> > >
> > > $$
> > > -\check{A}_c(s, a) - \lambda - \nu \log \pi^* + \nu \log \pi' + \nu = 0,
> > > $$
> > >
> > > $$
> > > \nu \log \pi^* = -\check{A}_c(s, a) + \nu \log \pi' - \lambda + \nu,
> > > $$
> > >
> > > $$
> > > \pi^* = \frac{1}{Z} \pi' \exp(-\mu \check{A}_c(s, a)),
> > > $$
> > >
> > > where $ \mu = \frac{1}{\nu} $ and $ Z $ is the normalizing constant to ensure $ \pi^* $ is a valid distribution. Thus, we have $ \pi^* \propto \pi' \exp(-\mu \check{A}_c(s, a)) $, which corresponds to the weights $ \exp(-\mu_2 \check{A}_c(s, a)) $ in our paper.
> > >
> > > For training the diffusion model, we consider two components (as discussed in [R1]):
> > >
> > > 1. Training the diffusion policy:
> > >
> > >    $$
> > >    \min\_\theta \mathbb{E}\_{t \sim \mathcal{U}([0,T]), z \sim \mathcal{N}(0, I), (s, a) \sim \mathcal{D}}
> > >    \left[ \\| z - z\_\theta(a_t, s, t) \\|\_2^2 \right],
> > >    $$
> > >
> > >    which serves as behavior cloning.
> > >
> > > 2. Optimizing the expected Q-values:
> > >
> > >    $$
> > >    \min\_\theta \mathbb{E}\_{s \sim \mathcal{D}, a \sim \pi\_\theta(a \vert s)}
> > >    \left[ \check{Q}\_c(s, a) \right].
> > >    $$
> > >
> > > As discussed in [R1] and proved in the FISOR paper, these two objectives can be combined into a single objective function:
> > >
> > > $$
> > > \min\_\theta \mathbb{E}\_{t \sim \mathcal{U}([0,T]), z \sim \mathcal{N}(0, 1), (s, a) \sim \mathcal{D}}
> > > \left[ -\mu \check{A}\_c(s, a) \cdot \\| z - z_\theta(a_t, s, t) \\|\_2^2 \right].
> > > $$
> > >
> > > However, as we discuss in our paper, we avoid behavior cloning for states in the unsafe region and aim to constrain updates to the current policy by limiting the distance from the previous **safe** target policy $ \pi' $. Thus, we optimize $ \check{A}_c(s, a) $ under the current policy instead of relying on $ (s, a) \sim \mathcal{D} $. By restricting the update distance from $ \pi' $, we define our objective function in Eq. (11) as:
> > >
> > > $$\min\_\theta \mathbb{E}\_{t \sim \mathcal{U}([0,T]), (s, a) \sim \pi\_\theta}
> > > \left[ \exp(-\mu\_2 \check{A}\_c(s, a)) \cdot \\| z\_\theta(a\_t, s, t) - z'(a\_t, s, t) \\|_2^2 \right],$$
> > >
> > > where $ z' $ represents the diffusion model under the target policy $ \pi' $, which has been trained with sufficient data and serves as a good approximation of $ z \sim \mathcal{N}(0, 1) $. We will provide more discussion in our final revision.
> > >
> > > [R1]: https://arxiv.org/pdf/2305.20081

---

> > > > ### Author Response · Authors · 2024-11-25
> > > > **Response to LCMx---New Response**
> > > >
> > > > >**Q3:**
> > > >
> > > > Although we simply set $ K_{\text{trust}} $ as half the total number of training episodes, it is already **safe** for the first collection. We copied our response to Yqdv here, which addresses your concerns:
> > > >
> > > > "In the following table, we report the average cost per episode (cost limit $ l = 10 $) based on $ 600,000 $ data samples with 600 trajectories for the original offline dataset, the first collection (after training for $ 200K $ steps), and the second collection (after training for $ 400K $ steps). We can clearly observe that the mean episode cost drops dramatically after each new collection, which implies that the policy is **safe enough** for exploration."
> > > >
> > > > |                         | **Mean Episode Cost** |
> > > > |-------------------------|-----------------------|
> > > > | Offline Dataset         | 69.72                 |
> > > > | First collected data    | 7.45                  |
> > > > | Second collected data   | 4.41                  |
> > > >
> > > > *Table: Average Episode Cost*
> > > >
> > > > Besides, we also highlight this information in the revised paper (Section 5.1 "Safe Offline RL with New Data Collection").

---

> ### Comment · Reviewer_LCMx · 2024-11-29
>
> Thanks for the authors' response. I am convinced by the authors and have no more concerns. Although other reviewers have some concerns, I believe this work can provide some insights into the offline safe RL field. After reading the authors' responses to other reviewers, I hold a positive attitude that they can address their problems to some extent. The author should declare their core contribution and clarify their motivation for designing the experiment to convince other reviewers. I tend to raise my score but I will first wait for other reviewers' responses.

---

> > ### Comment · Reviewer_LCMx · 2024-12-03
> >
> > Since many reviewers still have concerns, I believe this paper should be revised and resubmitted again. I recommend that the authors improve their work based on reviewers' comments.

---

### Author Response · Authors · 2024-11-21
**General Response**

We sincerely thank all the reviewers for their efforts during the review process and for providing constructive feedback. We provide our responses to the reviewers' comments below and have made major revisions to our revised manuscript. To enhance clarity, we have highlighted the revised text in blue for easy identification. Besides the specific response to each reviewer, we also list our responses to some general questions.

> **Difference between BARS and FISOR**

FISOR is a well-designed algorithm that inspired the design of our approach and demonstrates strong performance, particularly in hard constraint scenarios with good data coverage. However, our paper focuses on addressing the issue of using distribution regularization in offline safe RL (a topic neglected not only by FISOR). We differ from FISOR in several significant aspects, summarized below:
- Addressing Behavior Regularization Issues: This is the first paper to address how behavior regularization constraints in safe RL can lead to infeasible solutions, particularly with multiple constraints. Additionally, behavior regularization may cause the learned policy to imitate the behavior policy, even in states where the behavior policy performs poorly or is unsafe. This issue is not limited to FISOR; most existing algorithms fail to address this and cannot guarantee a safe policy when the behavior policy is suboptimal, as shown in Table 2 of our paper.
- Flexibility Beyond Hard Constraints: FISOR addresses only the "hard constraint" setting, where the cost value function represents the maximum constraint violation (largest state-wise cost) in the trajectory. This greatly limits the generalization of their approach to other scenarios where a cost limit is considered. Strictly controlling the hard constraint on certain dangerous states can also be achieved by adding a large penalty to the reward function for such states to avoid an unsafe policy. In contrast, BARS can handle a wide range of constraints, offering flexibility by adjusting the constraint limit in the algorithm.
- Applicability to Stochastic Systems: FISOR defines the feasible region using the value function for the hard constraint based on Hamilton-Jacobi (HJ) reachability from control theory. However, this formulation applies only to **deterministic systems**, addressing only a subset of RL problems, and cannot be generalized to problems with more flexible cost limits. The Bellman equation used in FISOR to train the value function no longer holds in stochastic systems.
- Optimizing for Safety and Reward: FISOR optimizes the policy by maximizing the value function to achieve the highest reward without considering constraints, which overlooks the fact that such a policy may not be safe. In contrast, we propose a "Region-Based Risk-Reward Optimization" in Eq. (8), which accounts for potential errors in value function estimation (likely in practice) and directly optimizes the policy based on the **current** policy. This approach enables BARS to avoid potential unsafe actions effectively.
---
> **Hyperparameter Concern**

As noted by Reviewers "LCMx," "Yqdv," and "WdBS," the selection of hyperparameters does contribute to our results. However, in our experiments presented in Table 1, we used the hyperparameters provided in Appendix, Table 7. For most tasks, we fixed the loss control terms at $b_1 = 1$ and $b_2 = 3$; only for the "AntVel" and "HalfCheetah" experiments did we adjust these values to $b_1 = 3$  and $b_2 = 1$, as these environments are less complex than others. In other words, the safety guarantees ensured by BARS are **not** due to parameter tuning. We did not perform extensive hyperparameter tuning; the strong results achieved are primarily attributed to our algorithm’s design.

Reviewer “WdBS” suggested that the same performance could be achieved by carefully tuning the hyperparameters of FISOR. We strongly disagree with this for the following reasons:
1. This statement could apply to any algorithm. The focus should be on addressing existing issues and limitations in offline safe RL rather than emphasizing parameter tuning.
2. Algorithms should be robust and use a unified set of parameters, whether in RL or safe RL. Designing an algorithm that is overly sensitive to hyperparameters is not practical.
3. FISOR cannot ensure safety when the behavior policy is suboptimal, as demonstrated in our paper. Even if safety could be achieved through hyperparameter tuning, this would not address the fundamental issue of behavior regularization.
4. The FISOR authors themselves state in their appendix (Section E), “However, we find that FISOR is not very sensitive to hyperparameter changes,” suggesting that parameter tuning would not significantly affect the results.
---
We hope that these clarifications address the common concerns and highlight the contributions of our work. We are committed to further refining the manuscript and incorporating any additional feedback.

---

### Author Response · Authors · 2024-11-26

Dear Reviewer,

We sincerely thank all the reviewers for their active engagement during the discussion phase.

We believe our work makes unique contributions to the field of offline safe RL, addressing the issue of behavior regularization— that has not been previously explored either theoretically or empirically. We have done our best to address every concern raised by the reviewers. As the author-reviewer discussion period is coming to an end, we are happy to respond to any further questions or comments you may have before the discussion period closes.

If our responses have satisfactorily resolved your concerns, we kindly ask you to consider raising the rating of our work. Thank you very much for your time and efforts!

---

### Meta-Review · Area_Chair_P4D2 · 2024-12-19

**Metareview:**

The paper proposes a novel algorithm, BARS, designed to address the issue of behavior regularization in offline safe reinforcement learning. The authors tackle the challenge of learning safe strategies in unsafe states by distinguishing between safe and unsafe states and applying region-based selective behavior regularization to optimize the policy. However, the primary weaknesses of this paper lie in its writing, particularly in the discussion of related work and the clarity of mathematical proofs, especially in differentiating itself from FISO. The authors need to better organize the paper to emphasize the novelty of their algorithm.

**Additional Comments On Reviewer Discussion:**

During a thorough review of the paper, the AC also noted inconsistencies in how references were cited and provided the following suggestions:
1. The ICLR main paper allows for 10 pages of content, sufficient for a conference submission. However, the authors relegated most of the discussion on related work to the appendix, creating significant challenges for readers, as pointed out by reviewers.
2. There is a lack of discussion on relevant papers in the field to clearly highlight the novelty of the proposed method. Reviewers repeatedly mentioned insufficient comparison with FISO.
3. References in lines 759–762 are particularly disorganized, combining a field-specific keyword with an excessive number of applications. This approach appears confusing to readers familiar with Safe RL. Furthermore, the paper fails to cite CPO (Constrained Policy Optimization) within this section, even though it may have been referenced elsewhere.
4. The Related Work section focuses exclusively on offline RL, which is insufficient. Reviewers suggested incorporating additional analysis from the perspective of safe control.
5. Equations are not followed by punctuation marks.
6. The authors utilize Safety-Gymnasium, a benchmark for safe RL, but fail to cite the original paper, Safety-Gymnasium: A Unified Safe Reinforcement Learning Benchmark.

To ensure an informed decision, the AC dedicated significant time to thoroughly reviewing the manuscript. Considering the paper’s scores, which are significantly below the acceptance threshold, and the conclusions drawn from the AC’s review, the paper has been rejected.

---

### Decision · Program_Chairs · 2025-01-22

Reject